# Unified Screening for Multiple Diseases

**Yiğit Narter** [1]   **Alihan Hüyük** [2]   **Mihaela van der Schaar** [3]   **Cem Tekin** [1]

## Abstract

Current screening programs that focus on improving patient health while minimizing screening costs are tailored for individual diseases. Designing unified screening programs for multiple diseases requires carefully balancing competing disease risks, which is an open problem. In this work, we address this problem by casting unified screening as a referral problem, in which we choose to activate a subset of screening policies for individual diseases by accounting for competing risks that influence patient outcomes. We introduce a novel optimization framework that incorporates disease risks, budget constraints, and diagnostic error limits and characterize the structural properties of the optimal referral policy. For the unified screening of two diseases, we show that the optimal activation threshold for the screening of one disease depends on the risk of the other, resulting in decision boundaries with distinct risk-dependent profiles. We compare our unified model with independent screening programs that apply isolated activation thresholds for screening of each disease. Our approach optimizes screening decisions collectively, improving overall survival outcomes, particularly for patients with high disease risks.

## 1. Introduction

The challenge of designing effective screening programs for disease detection is central to improving healthcare outcomes. Traditionally, screening programs are developed in isolation for individual diseases, ignoring the interactions between diseases and risks for different conditions

[1]Department of Electrical and Electronics Engineering, Bilkent University, Ankara, Turkey [2]Department of Computer Science, Harvard University, Cambridge, MA, US [3]University of Cambridge, Cambridge, UK. Correspondence to: Yiğit Narter <yigit.narter@ug.bilkent.edu.tr>, Cem Tekin <cemtekin@ee.bilkent.edu.tr>.

*Proceedings of the 42nd International Conference on Machine Learning*, Vancouver, Canada. PMLR 267, 2025. Copyright 2025 by the author(s).

(De Mutsert et al., 2009; Weiner et al., 2006; Thayer et al., 2010). This limitation reduces the efficiency of healthcare interventions, as potential synergies between diseases are overlooked, and resources may be misallocated. Screening a patient with high risks for two conditions for only one condition might fail to improve their overall health outcomes, whereas a unified screening approach could yield better results (Fan et al., 2024; Huang et al., 2023; Yang et al., 2006) by addressing the competing risks, which refer to events that interfere with or alter the likelihood of observing the event of interest (Pintilie, 2011).

For instance, consider a patient who has both cardiovascular disease and cancer but is only screened for cancer, where the adverse event from cancer happens to occur after the adverse event from heart disease. When screening for cancer, if the patient unexpectedly suffers from heart disease, then the cancer screening offers no benefit to the patient in terms of quality-adjusted life years (QALYs). However, if the patient is screened for both diseases, heart disease will be identified earlier. In that case, the adverse event due to heart disease will be prevented, and cancer screening will result in QALY gain by detecting cancer before the adverse event associated with it happens. However, screening the patient for both diseases is costly. Hence, optimal allocation of screening resources to patients with different risk profiles is an important task.

In this paper, we propose a novel framework for designing unified screening programs that simultaneously address multiple diseases. Unlike existing programs, our approach accounts for the competing risks associated with each disease when determining whom to screen for which condition while also considering resource constraints, an important factor in screening programs (Cevik et al., 2018; Bansal et al., 2020; Teh et al., 2015). We introduce an optimization formulation for the screening problem, which incorporates budget constraints and diagnostic error limits. Using properties of the Lagrange dual function, we characterize optimal decision boundaries and validate our findings through extensive in-silico experiments. In particular, for the case of unified screening of two diseases, we show that the optimal activation threshold for one disease is not static but a function of the risk of the other disease. Instead of a single threshold per disease, our model introduces multiple regions that correspond to distinct combinations of screen-

ing activations. These regions emerge due to the interplay of competing risks, where the decision to screen for one disease depends not only on its individual risk but also on its interaction with the risks of other diseases. This also results in decision boundaries with a curvy shape, which reflects the interplay between the risks of the diseases.

Our experimental results demonstrate that the mathematical characterizations of the optimal policies align closely with the numerical solution of the convex program (see Figure 1). Importantly, we compare the performance of our unified screening model against that of independent screening programs. Independent screening programs operate under the assumption that diseases are unrelated and apply distinct threshold policies for each condition. Specifically, they activate screening for a disease if the patient's risk for that disease exceeds a predetermined threshold, independent of the risks for other conditions. In contrast, our unified model accounts for the interplay between diseases and optimizes the screening policies collectively rather than in isolation. This approach enhances survival times for patients whose risk profiles indicate significant interactions between diseases, particularly in scenarios where one disease's optimal threshold influences the other's. For instance, in cases of competing risks (Wolbers et al., 2014; Pintilie, 2011; Allignol et al., 2011), addressing one condition indirectly benefits the other, while in cases of individual risks, prioritizing the most pressing condition ensures optimal resource allocation. Our results show that, when compared with the independent screening model, the unified model prioritizes patients with higher relative risks for a specific disease, activating screenings for these individuals in cases where independent screening would not (as shown by the colored regions in Figure 1(c)). Conversely, it reduces screenings for patients with moderate risks for both diseases, resulting in grey regions in Figure 1(c). In essence, the model places greater emphasis on addressing higher relative risks, which leads to better health outcomes for these patients (see Figure 2).

Our unified screening framework represents a significant step forward in improving the design and efficiency of screening programs, ensuring better health outcomes in resource-constrained environments. In silico experiments for the unified screening of two diseases demonstrate that survival times improve compared to the optimal individual screening programs, validating the benefit of unified screening (see Figure 2).

## 2. Related Work

From a medical perspective, prior research has predominantly focused on optimizing sampling schedules or decision-making processes for single-disease screenings. These strategies often aim to determine or evaluate inclusion criteria (Tomaszewski et al., 2022; Bauer et al., 2015;

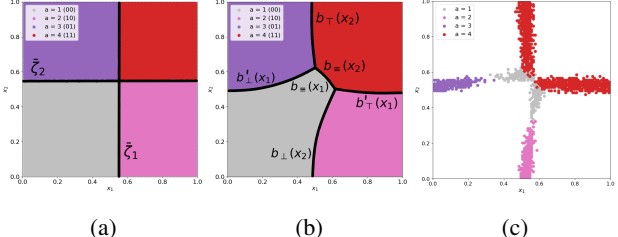

(a)  (b)  (c)

Figure 1: Optimal referral policy for $N = 2$ under uniform risk distribution for (a) independent screening model, (b) our unified screening model. Grey: $a = 1$ (no screening), Pink: $a = 2$ (screening only the first disease), Purple: $a = 3$ (screening only the second disease), Red: $a = 4$ (screening both diseases). Notice that unlike in the independent case with constant thresholds, the activation threshold for the first disease (to the pink and red regions) is a curve which is in fact a function of $x_2$, the risk of the other disease. The difference between two policies is given in (c).

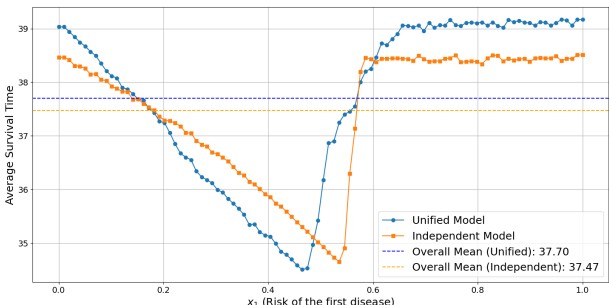

Figure 2: Average survival times (in years excess of 50) with respect to $x_1$ (the first disease risk) for both the unified and independent screening programs computed by taking the mean survival times over all $x_2$ values (the second disease risk), in the case of screening for two diseases, along with the mean survival times for both programs indicated by the horizontal dashed lines. Details are provided in Section 5.

Fernandez et al., 1991; Yu et al., 2021; Xiao et al., 2022; Idrees et al., 2021; Wang et al., 2024b) and screening policy frequency (Hemmati et al., 2024; Wu & Suen, 2022; Wang et al., 2024b; Wu et al., 2024). Most of the existing studies focus on Markov models and processes (Wu et al., 2024; Wu & Suen, 2022; Zhang et al., 2012a;b; Maillart et al., 2008). Kamalzadeh et al. (2021) propose a framework combining partially observable Markov Decision Processes (POMDP), Hidden Markov Models (HMM), and Predictive Risk Modeling (PRM) to design individualized diabetes screening policies. To optimize decision making in healthcare, Bertsimas et al. (2018) propose a framework for prostate cancer screening that identifies strategies effective across multiple mathematical models using heuristic search.

Cost-effectiveness analyses have also been a critical component of the literature to identify the optimal trade-off between screening costs and health benefits (Xia et al., 2024; Qin et al., 2022; 2024; Wang et al., 2024a; Bao et al., 2022).

Table 1: Comparison of Related Works.

| Reference | Joint Screening | Multiple Diseases | Resource Constraints | Competing Risks |
|---|---|---|---|---|
| Yala et al. (2022) | No | No | Yes | No |
| Cevik et al. (2018) | No | No | Yes | No |
| Huang et al. (2023) | Yes | Yes | No | No |
| Wright et al. (2015) | No | No | No | Yes |
| Peng & Xiang (2021) | Yes | No | No | Yes |
| Alaa & Van Der Schaar (2016) | No | No | Yes | No |
| **This Work** | Yes | Yes | Yes | Yes |

Resource constraints, such as limited budgets, personnel, and equipment (Yadla et al., 2024; Khoza-Shangase et al., 2017), play a fundamental role in these analyses, as they directly impact the feasibility and scalability of screening programs (Cevik et al., 2018; Bansal et al., 2020). Cevik et al. (2018) tackle resource constraints by proposing a constrained POMDP model to optimally allocate limited mammography screening resources, and prioritize patients based on their risk levels and screening capacity. Yala et al. (2022) present a reinforcement learning-based framework, which optimizes resource-constrained breast cancer screening by AI-driven risk models, to enhance early detection and minimize excessive screening. However, to the best of our knowledge, none of the existing methods address both resource constraints and competing risks of multiple diseases simultaneously, which leaves a gap in optimizing multi-disease screening programs under such conditions.

A competing risks problem in our approach refers to the assessment of the likelihood of health benefits from an intervention, considering the presence of other potential outcomes that may interfere with or prevent the occurrence of the event of interest (Varadhan et al., 2010). This approach enables the analysis of both the timing of the first observed event and its type (Wolbers et al., 2014). Competing risk analyses have been tackled in the literature (Allignol et al., 2011; Wolbers et al., 2014; Satagopan et al., 2004; Pintilie, 2011; Cho et al., 2022), which highlights its importance. Stenling et al. (2020) employ a competing risks approach to develop lifetime risk models for cardiovascular events, while Wright et al. (2015) develop a competing risks model for screening preeclampsia by incorporating maternal demographic characteristics and medical history. In a semi-competing risks framework, Peng & Xiang (2021) propose a joint feature screening method that is based on the correlation ranking of gene features, to address challenges in ultrahigh-dimensional breast cancer data analysis.

Several studies highlight the benefits of joint screening programs. Fan et al. (2024) evaluate joint screening for prostate, lung, colorectal, and ovarian cancer, and demonstrate that joint cancer screening reduces all-cause and all-neoplasm mortality, which makes it a promising strategy. Huang et al. (2023) discuss that combined cancer screening would enhance cost-effectiveness, and improve early detection through shared risk factors. Regarding the screening for multiple diseases, Yang et al. (2006) evaluate the integration of colorectal cancer (CRC) screening into a multi-disease program through a Markov model. Their study projects that multiple disease screening reduces CRC mortality by 33%, which outperforms single-disease screening (23% reduction). Huang et al. (2023) implement a combined cancer screening program for colorectal, breast, lung, liver, and stomach cancers with C-BLAST trial, using a multi-stage design.

Our work on designing disease screening programs, which is about making decisions based on accumulating evidence to optimize outcomes under uncertainty, is closely related to sequential hypothesis testing (Chernoff, 1992; Drugowitsch et al., 2014; 2012; Shvartsman et al., 2015). With multiple diseases, the general problem is an instance of sequential hypothesis testing with multiple hypotheses. Prior work, such as by Alaa & Van Der Schaar (2016) examines resource allocation with a Bayesian model in single-hypothesis (disease) scenarios but does not extend to the complexities of multi-disease contexts. However, in our context, multiple diseases correspond to multiple hypotheses, which transforms our problem into an active learning challenge, where actions influence the acquisition of information across diseases.

Different than all of the works listed, the framework developed in this paper integrates the optimization of referral decisions with probabilistic decision-making. The proof techniques illustrate how boundaries in the decision space can be derived, connecting to geometric approaches found in threshold-based medical screening policies. By incorporating stochastic constraints, our approach generalizes these methods into a probabilistic framework that adaptively balances performance metrics, such as survival time and cost. Table 1 compares our work with the closely related works. See Appendix A for more detail on related works.

# 3. Problem Formulation

## 3.1. The Unified Screening Problem

**Diseases and Adverse Events.** We consider the screening of $N$ diseases. Patients have latent disease states represented by random variables $\Theta_n \in \{0, 1\}$, $n \in [N]$. The disease state vector is defined as $\Theta = [\Theta_1, \ldots, \Theta_N]^\mathsf{T}$. If $\Theta_n = 1$, the patient experiences an adverse event related to disease $n$ at a random time $T_n \in \mathbb{N}_+$.[1] If $\Theta_n = 0$, no such event occurs, and $T_n = \infty$. The pmf for $T_n$ conditioned on $\Theta_n$ is given as

$$
p_{T_n|\Theta_n}(t \mid \theta_n) = \begin{cases} 1, & \text{if } \theta_n = 0 \text{ and } t = \infty, \\ f_{T_n}(t), & \text{if } \theta_n = 1 \text{ and } t \in \mathbb{N}_+, \\ 0, & \text{otherwise.} \end{cases}
$$

Here, $f_{T_n}(t)$ is a probability distribution (e.g., geometric) defined on $\mathbb{N}_+$. The CDF of $T_n$ given $\Theta_n = \theta_n$ is $F_{T_n|\Theta_n}(t|\theta_n) = \mathbb{P}(T_n \leq t|\Theta_n = \theta_n)$. The random vector $T = [T_1, \ldots, T_N]^\mathsf{T}$ represents the adverse event times for all diseases.

**Risks and Screening Targets.** Each patient has a risk vector $X = [X_1, \ldots, X_N]^\mathsf{T} \in [0, 1]^N$, where $X_i$ denotes the prior probability of the patient having disease $i$.[2] For $n \in [N]$:

$$
\mathbb{P}\{\Theta_n = \theta_n \mid X = x\} = \begin{cases} x_n, & \text{if } \theta_n = 1, \\ 1 - x_n, & \text{if } \theta_n = 0. \end{cases}
$$

The distribution of risk vector over patient population is given by $p_X(x)$. There are $N$ screening targets, each of which is associated with one disease. The $n$th screening target $Y_n : \mathbb{R}_+ \to \mathbb{R}$ is a random function with $Y_n(t)$ representing its value at time $t$. The conditional distribution of $Y_n(t)$ given $\Theta_n$ is $p_{Y_n(t)|\Theta_n}(y_n|\theta_n)$.[3] The vector of random target functions is represented as $Y = [Y_1, \ldots, Y_N]^\mathsf{T}$.

**Probabilistic Dependencies.** The random elements in the system can be represented as a tuple $(X, \Theta, T, Y)$. The probabilistic relationships between these random elements are encoded in the Bayesian network given in Figure 3, which implies the following factorization for the joint distribution of $(X, \Theta, T, Y)$: $p_{X,\Theta,T,Y}(x, \theta, t, y) = p_X(x)p_{\Theta|X}(\theta \mid x)p_{Y|\Theta}(y \mid \theta)p_{T|\Theta,Y}(t \mid \theta, y)$. Moreover, the following conditional independence statements hold as a result of d-separation: $p_{\Theta|X}(\theta|x) = \prod_{n=1}^N p_{\Theta_n|X}(\theta_n|x)$, $p_{T|\Theta}(t|\theta) =$

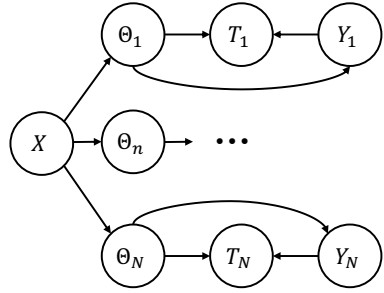

Figure 3: Bayesian network of $(X, \Theta, T, Y)$.

$\prod_{n=1}^N p_{T_n|\Theta_n}(t_n|\theta_n)$, $p_{T|X}(t|x) = \prod_{n=1}^N p_{T_n|X}(t_n|x)$, $p_{T|X,\Theta}(t|x,\theta) = p_{T|\Theta}(t|\theta) = \prod_{n=1}^N p_{T_n|\Theta_n}(t_n|\theta_n)$, $p_{Y_n(t)|X,\Theta}(y_n) = p_{Y_n(t)|\Theta_n}(y_n)$. In addition, we assume that given $\Theta$, screening samples taken at different times are independent: $Y(t) \perp\!\!\!\perp Y(t') \mid \Theta$ for $t' \neq t$.

**Screening and Diagnostic Actions.** For each patient, the decision-maker first observers the risk vector $x$. Then, it needs to determine the disease states $\Theta_n$ by observing costly samples of the screening targets $Y_n$ before the adverse events associated with the diseases occur. At each $t$, they take two types of actions. The first is the *screening action* $\delta(x, t) \in \{0, 1\}^N$. If $\delta_n(x, t) = 1$, then the decision-maker observes the $n$th screening target $Y_n(t)$. If $\delta_n(x, t) = 0$, then they do not observe the $n$th screening target. We denote with $\#_n(x, T) = |\{t \in [0, T) : \delta_n(x, t) = 1\}|$ the total number of observations taken from the $n$th screening target up to time $T$. The second is the *diagnostic action* $\hat{\theta}(x, t) \in \{0, 1\}^N$. The first time point when $\hat{\theta}_n(x, t) = 1$ indicates that the decision-maker has diagnosed the patient with the $n$th disease. We denote with $t_n^*(x) = \min\{t \in \mathbb{R}_+ : \hat{\theta}_n(x, t) = 1\}$ the diagnosis time for the $n$th disease for patient with risk vector $x$. When the dependence on $x$ is clear from the context, we will drop the argument inside the paranthesis from the notation, and use $\delta_n(t), \hat{\theta}_n(t), t_n^*$, etc.

**Decision-making Policies.** The decision-maker takes screening actions $\delta$ and diagnostic actions $\hat{\theta}$ according to some screening policy $\delta^\pi$ and some diagnostic policy $\hat{\theta}^\pi$ respectively, which constitutes a complete policy $\pi = (\delta^\pi, \hat{\theta}^\pi)$. Denote with $\mathcal{I}_\delta(T) = \{X\} \cup \{Y_n(t)\}_{n \in [N], t \in [0,T): \delta_n(t)=1} \cup \{\delta(t)\}_{t \in [0,T)} \cup \{\hat{\theta}(t)\}_{t \in [0,T)}$ all of the information available at time $T$ to the screening policy and $\mathcal{I}_{\hat{\theta}}(T) = \mathcal{I}_\delta(T) \cup \{\delta(T)\} \cup \{Y_n(T)\}_{n \in [N]: \delta_n(T)=1}$ the diagnostic policy. Clearly, $\delta(t) = \delta^\pi(t) = \delta^\pi(t, \mathcal{I}_\delta(t))$ and $\hat{\theta}(t) = \hat{\theta}^\pi(t) = \hat{\theta}^\pi(t, \mathcal{I}_{\hat{\theta}}(t))$. Let $\Pi$ represent the set of all joint screening and diagnostic policies.

**Objective.** Each sample from the screening target $n \in [N]$ has a fixed cost of $c_n \in \mathbb{R}_+$. For instance, $c_n$ can represent

---

[1] A time unit can correspond to a day, month, year, etc.

[2] One can use off-the-shelf risk prediction models to obtain disease risks for particular diseases. For instance, the Gail model (for breast cancer), QRISK3 (for cardiovascular disease), (normalized) polygenic risk scores, and even AI-based models can be used to provide risk scores.

[3] $Y_n$ may represent a biomarker value, clinical metric, or similar quantity related to disease $n$. The randomness in $Y_n$ can be attributed to measurement noise or natural biological fluctuations.

the cost of mammography or chest X-ray. We say the $n$th disease is diagnosed on time if and only if (iff)

$$\mathcal{O}_n = \{\Theta_n = 1\} \wedge \{t_n^* < T_n\}.$$

Let $T^*$ be the time point when the first adverse event that is not diagnosed on time occurs such that $T^* = \min\{T_n : n \in [N], \neg\mathcal{O}_n\}$. Then, the objective of the decision-maker is to maximize $T^*$ up to some survival time $T_0 \in \mathbb{N}_+$ subject to constraints on budget and diagnostic error rates, which is formalized as a joint screening and diagnosis problem:

$$\begin{aligned}
\text{maximize} \quad & \mathbb{E}[\min\{T_0, T^*\}] \quad (1)\\
\text{s.t.} \quad & \mathbb{E}[\textstyle\sum_{n\in[N]} c_n \times \#_n(\min\{T_0, T_n, t_n^*\})] \le B,\\
& \mathbb{P}\{\Theta_n = 0, \hat{\theta}_n(t) = 1\} \le \alpha_n, \forall t \in [T_0], n \in [N],\\
& \pi \in \Pi,
\end{aligned}$$

where $B$ is the total screening budget and $\alpha_n$ is the maximum allowed false positive rate for disease $n$. The expectations are taken over the randomness induced by interaction of $\pi$ with $(X, \Theta, Y, T)$.

### 3.2. The Referral Problem

The referral problem is about determining the optimal screening and diagnostic policy to use under each risk vector given a set of existing policies, rather than creating a completely new policy from scratch. This is crucial because the joint optimization approach in (1) can yield complex policies, which are difficult to implement in real-world clinical practice. Moreover, new policies are hard to justify in clinical guidelines, especially if they are entirely novel, and implementing novel clinical guidelines is another challenge due to many factors, including legislation (Wang et al., 2023; Francke et al., 2008; Correa et al., 2020). On the other hand, combining existing policies in an optimal way, as we do by solving the referral problem, is feasible and easier to support in practice (Stoll & Norton, 2018). Our approach, which optimizes the referral process, provides a lower bound on the performance of (1).

Suppose an individual policy $\pi_n = (\delta^{\pi_n}, \hat{\theta}^{\pi_n})$ for each disease $n \in [N]$ is already provided. Each of these policies only ever diagnose their corresponding diseases such that $\hat{\theta}_{n'}^{\pi_n}(x, t) = 0$ if $n' \ne n$ for all $x \in [0, 1]^N$ and $t \in \mathbb{R}_+$. To represent which policies are active for a given risk vector $x$, define the referral decision $\rho : [0, 1]^N \to \{0, 1\}^N$ such that $\rho(x) = [\rho_1(x), \rho_2(x), \dots, \rho_N(x)]^\mathsf{T}$, where

$$\rho_n(x) = \begin{cases} 1 & \text{if the policy for disease } n \text{ is active,} \\ 0 & \text{otherwise.} \end{cases}$$

This leads to the following composite policy:

$$\delta_n(x, t) = \begin{cases} 1 & \text{if } \rho_n(x) = 1 \wedge \delta_n^{\pi_n}(x, t) = 1, \\ 0 & \text{otherwise.} \end{cases}$$

$$\hat{\theta}_n(x, t) = \begin{cases} 1 & \text{if } \rho_n(x) = 1 \wedge \hat{\theta}_n^{\pi_n}(x, t) = 1, \\ 0 & \text{otherwise.} \end{cases}$$

This composite policy samples the $n$th target when screening for disease $n$ is active and the policy for that disease asks to screen for that target.

Let $\Pi^{\text{ref}} \subset \Pi$ represent the set of all referral decisions given individual screening and diagnostic policies for all diseases. As a subset of the referral decisions, we define the *independent* referral decisions as $\bar{\rho}(X) = [\bar{\rho}_1(x_1), \bar{\rho}_2(x_2), \dots, \bar{\rho}_N(x_N)]^\mathsf{T}$, where $\bar{\rho}_n$'s are independent from $x_i, \forall i \ne n$. In this case, unlike the general referral decisions where the activation of a disease screening depends on the risk vector $x$, in the independent referral decision, the activation of the $n$th disease depends on only $x_n$. Let $\Pi^{\text{ind}} \subset \Pi^{\text{ref}}$ denote the set of independent referral decisions.

To represent all possible combinations of screening decisions for $N$ diseases as integers, we define the action $a(\rho) : \{0, 1\}^N \to \mathcal{A}$, where $\mathcal{A} = \{1, 2, \dots, 2^N\}$, mapping referral decisions $\rho$ to integer indices. In the case of $N = 2$, the actions $a = 1, 2, 3, 4$, correspond to the following referral decisions respectively: nothing is screened, only the first disease is screened, only the second disease is screened, and both diseases are screened. Lastly, we define $q(a|x)$ as the probability of selecting action $a$ (and consequently its corresponding referral decision $\rho$), given a patient's risk vector $x$. Hence for any $x$, we have $q(a|x) \ge 0, \forall a \in \mathcal{A}$ and $\sum_{a \in \mathcal{A}} q(a|x) = 1$.

We formalize the problem of finding the optimal referral decision $\rho^*$, which provides a lower bound on the optimal value of (1) as follows:[4]

$$\text{maximize}_q \int_{x \in \mathcal{X}} p_X(x) \left( \sum_{a \in \mathcal{A}} q(a|x) \cdot r_{a,x} \right) dx \quad (2)$$

---

[4]If the goal were to learn the optimal referral policy directly from a dataset of patient screening trajectories, the problem could indeed be framed as a policy learning task with an appropriately defined cost function. However, our work takes a complementary, orthogonal approach. We begin with a well-defined optimization problem and focus on analytically characterizing the structure of its optimal solution. Rather than learning from data, our emphasis is on understanding the geometry and properties of the optimal referral rule under a known probabilistic model.

$$
\text{s.t.} \begin{cases}
\displaystyle\int_{x\in\mathcal{X}} p_X(x) \left( \sum_{a\in\mathcal{A}} q(a|x) \cdot m_{a,x} \right) dx \leq B, \\[4pt]
q(a|x) \geq 0, \quad \forall a \in \mathcal{A}, \forall x \in \mathcal{X}, \\[4pt]
\displaystyle\sum_{a\in\mathcal{A}} q(a|x) = 1, \quad \forall x \in \mathcal{X}, \\[4pt]
\mathbb{P}\{\Theta_n = 0, \hat{\theta}_n(t) = 1\} \leq \alpha_n, \forall t \in [T_0], n \in [N]
\end{cases}
$$

where $r_{a,x} := \mathbb{E}[\min\{T_0, T^*\}|a, x]$, $m_{a,x} := \sum_n c_n \mathbb{E}[\#_n | a, x]$ and $\mathcal{A}$ is the set of all actions.

## 4. The Optimal Referral Decision

In order to characterize the structure of the optimal referral decision, we focus on the case of $N = 2$, representing two diseases and two screening targets corresponding to each disease. This choice is motivated by the fact that screening for two diseases would be common and easier to adapt in clinical practice, whereas screening for three or more diseases simultaneously is much less frequent and often impractical (Ruhwald et al., 2022; Davis et al., 2015; Deverka et al., 2022; Castle, 2022). Additionally, the case of $N = 2$ allows for a clear visual inspection of the results and provides insights that can be intuitively generalized to higher dimensions. On the other hand, characterizing the optimal referral policy becomes significantly more complex and far less intuitive when considering more than two diseases. One would expect the number of boundaries of interest (e.g., as shown in Figure 1(b)) to grow combinatorial with $N$, as one must account for boundaries that separate screening decisions across different subsets of diseases. We assume that $p_X$ is a uniform distribution over the risk vector domain $[0, 1]^2$.

We restrict our attention to deterministic screening policies where the screening frequency remains constant over time. This includes periodic screening as a special case. This choice is supported by the literature (Curry et al., 2018; Wolf et al., 2024; Canadian Task Force on Preventive Health Care, 2011; Eddy, 1989; Bhatia et al., 2022; Shapiro, 1992), for example annual screening (Perros et al., 1995; Nagao & Warnakulasuriya, 2003), which highlights that periodic screening is (or should be) widely adopted in practice due to its health outcomes, simplicity and ease of implementation. Adaptive screening policies, while potentially more flexible, are often more complex and require continuous monitoring and adjustments, which may not be feasible or cost-effective in many clinical settings.

For each disease $n \in \{1, 2\}$, the screening policy provided is to take samples from the $n$th target (one target corresponding to each disease) with a fixed schedule until the patient is diagnosed with the disease (or an adverse event occurs, interrupting screening). Let $\{\tau_{n1}, \ldots, \tau_{nS_n}\}$ denote the sampling schedule for the $n$th disease, where $\tau_{ni}$ is the

time point that the $i$th sample is taken ($\tau_{n1} = 0$), and $S_n$ is the maximum sampling number. Since we do not assume an adaptive scheduling, $S_n$ is deterministic. We have:

$$
\delta_n^{\pi_n}(t) = \begin{cases}
1 \text{ if } E_n(t) \wedge \forall t' \in [0, t), \hat{\theta}_n^{\pi_n}(t') = 0, \\
0 \text{ otherwise}
\end{cases} \tag{3}
$$

where $E_n(t) := \exists i \in [S_n] : t = \tau_{ni}$.

The diagnostic policy is to issue a diagnosis only if the posterior probability of a patient having the disease exceeds some threshold $\gamma_n \in (0, 1)$ such that

$$
\hat{\theta}_n^{\pi_n}(t) = \begin{cases}
1 & \text{if } \dfrac{x_n}{1 - x_n} \cdot LR_n(t) \geq \dfrac{\gamma_n}{1 - \gamma_n}, \\
0 & \text{otherwise}
\end{cases} \tag{4}
$$

where $LR_n(t) = \prod_{i:\tau_{ni}\in[0,t)} \frac{p_{Y_n(\tau_{ni})|\Theta_n=1}(y_n(\tau_{ni}))}{p_{Y_n(\tau_{ni})|\Theta_n=0}(y_n(\tau_{ni}))}$. For this to be well defined, assume $p_{Y_n|\Theta_n=1}$ is absolutely continuous with respect to $p_{Y_n|\Theta_n=0}$. In this model, the diagnostic decision is based on the disease probability reaching a certain level. Although it is hard to create a model, given the difficulty of capturing all the factors clinicians consider, a probabilistic approach is applicable for a diagnosis decision (Agency for Healthcare Research and Quality, 2022; Sox et al., 1989). This model maintains the false positive rate below a specific level, and an essential aspect of this diagnosis policy is that patients are never diagnosed as "disease-free" given how there may exist medical conditions that may remain undiagnosed (National Institutes of Health (NIH), 2023; Tifft & Adams, 2014) due to their difficulty in being identified (Macnamara et al., 2020; Global Genes, 2023). Once screening begins, deterministic sampling continues until either the disease is detected or an adverse event related to the disease occurs.

First, note that the error constraint in (2) is satisfied for a suitable choice of $\gamma_n$. We can rewrite (2) as:

$$
\text{maximize}_q \int_{x\in\mathcal{X}} \left( \sum_{a=1}^{4} q(a|x) r_{a,x} \right) dx \tag{5}
$$

$$
\text{s.t.} \begin{cases}
\displaystyle\int_{x\in\mathcal{X}} \left( \sum_{a=1}^{4} q(a|x) m_{a,x} \right) dx \leq B, \\[8pt]
q(a|x) \geq 0, \forall a \in [4], x \in \mathcal{X}, \displaystyle\sum_{a=1}^{4} q(a|x) = 1, \forall x \in \mathcal{X}.
\end{cases}
$$

Let $p^*$ represent the optimal value of this problem. The Lagrange dual function associated with the above problem takes the form

$$
g(\lambda) = \max_q \int_{x\in\mathcal{X}} \left( \sum_{a=1}^{4} q(a|x)(r_{a,x} - \lambda(m_{a,x} - B)) \right) dx
$$

$$\text{s.t. } q(a|x) \geq 0, \sum_{a=1}^{4} q(a|x) = 1, \forall x \in \mathcal{X}, a \in [4], \qquad (6)$$

where $\lambda \geq 0$. The Lagrange dual function provides an upper bound for the problem, i.e., $g(\lambda) \geq p^*$. Given $x$, we pay attention to the term $\sum_{a=1}^{4} q(a|x)\kappa_a(x)$ in (6), where $\kappa_a(x) := r_{a,x} - \lambda m_{a,x}$. This term is maximized when $q(a^*|x) = 1$ for some $a^* \in \arg\max_{a \in \mathcal{A}} \kappa_a(x)$. The Lagrange dual problem associated with (5) takes the following form: minimize $g(\lambda)$ s.t. $\lambda \geq 0$. Let $\lambda^*$ represent the dual optimum. Since (5) is a feasible LP, strong duality holds. Therefore, $g(\lambda^*) = p^*$. This implies that the optimal policy for (5) takes the form $a^*(x) \in \arg\max_{a \in [4]} r_{a,x} - \lambda^* m_{a,x}$, $x \in \mathcal{X}$.

To determine the optimal referral decision, we will consider the referral decision $\rho_1^*(x)$ of the first disease. Intuitively, $\rho_1^*(x) = 1$ for $x$'s such that $\kappa$'s associated with actions where the first disease is screened ($a = 2$ or $a = 4$) exceed the ones associated with actions where the first disease is not screened ($a = 1$ or $a = 3$). A similar intuition holds for $\rho_2^*(x)$ as well. Define sub-rule $\rho_{i,j}(x) := \mathbb{1}\{\kappa_i(x) > \kappa_j(x)\}$. The following proposition characterizes $\rho_1^*(x)$ in terms of sub-rules $\rho_{i,j}(x)$s.

**Proposition 4.1.** $\rho_1^*(x) = \rho_{2,1}(x) \vee (\rho_{4,1}(x) \wedge \rho_{4,3}(x))$ and $\rho_2^*(x) = \rho_{3,1}(x) \vee (\rho_{4,1}(x) \wedge \rho_{4,2}(x))$.

Proposition 4.1 gives a complete characterization of when to screen for diseases 1 and 2. Next, we characterize a special case when each sub-rule in Proposition 4.1 takes an intuitive form. Given $x_2 \in [0,1]$, for any $(i,j)$, let

$$b_{i,j}(x_2) := \begin{cases} +\infty, & \text{if } \kappa_i(x) - \kappa_j(x) < 0, \forall x_1 \in [0,1] \\ -\infty, & \text{if } \kappa_i(x) - \kappa_j(x) > 0, \forall x_1 \in [0,1] \\ \{x_1 \in [0,1] : \kappa_i(x) - \kappa_j(x) = 0\}, & \text{else.} \end{cases}$$

The following three lemmas characterize the forms of the decision boundaries $b_\perp, b_\equiv, b_\top : [0,1] \to [0,1] \cup \{-\infty, +\infty\}$ that correspond to sub-rules $\rho_{2,1}, \rho_{4,1}$, and $\rho_{4,3}$ used to define $\rho_1^*(x)$.[5] The form of the decision boundaries for the sub-rules of $\rho_2^*(x)$ follows from the symmetry.

**Lemma 4.2.** For each $x_2 \in [0,1]$, there is a unique $b_\perp(x_2) \in (0,1)$. Thus, $\rho_{2,1}(x) = \mathbb{1}\{x_1 > b_\perp(x_2)\}$. Moreover, $b_\perp$ is an increasing function, i.e., $b_\perp(x_2) < b_\perp(x_2')$ if $x_2 < x_2'$.

**Lemma 4.3.** Assume that if a root of $\kappa_4(x) - \kappa_1(x)$ exists for a given $x_2 \in [0,1]$, then it is unique. $b_\perp(0) < b_\equiv(0)$, $b_\perp(1) > b_\equiv(1)$. Thus, $\rho_{4,1}(x) = \mathbb{1}\{x_1 > b_\equiv(x_2)\}$. If $b_\equiv(x_2)$ is decreasing in $x_2$, then, $b_\perp(x_2)$ and $b_\equiv(x_2)$ have a unique intercept $x_{2\perp} \in (0,1)$.

**Lemma 4.4.** For each $x_2 \in [0,1]$, there is a unique $b_\top(x_2) \in (0,1)$. Thus, $\rho_{4,3}(x) = \mathbb{1}\{x_1 > b_\top(x_2)\}$. In

---
[5]Technical assumptions and proofs are given in the Appendix.

addition, $b_\top$ satisfy the following properties: (i) $b_\perp(0) \leq b_\top(x_2) \leq b_\perp(x_2)$ for all $x_2 \in [0,1]$; (ii) $b_\perp(0) = b_\top(0) = b_\top(1)$; (iii) $b_\top(0) = b_\top(1) \leq b_\top(x_2)$ for all $x_2 \in [0,1]$.

Combining the lemmata above we characterize the form of the optimal referral decision as follows.

**Proposition 4.5.** When $b_\equiv(x_2)$ is decreasing, and $b_\equiv(x_2)$ and $b_\top(x_2)$ have a unique intercept $x_{2\top}$, then the optimal referral decision $\rho_1^*(x)$ has the form $\rho_1^*(x) = \mathbb{1}\{x_1 > \zeta_1(x_2)\}$ where

$$\zeta_1(x_2) = \begin{cases} b_\perp(x_2) & \text{if} \quad x_2 \in [0, x_{2\perp}] \\ b_\equiv(x_2) & \text{if} \quad x_2 \in (x_{2\perp}, x_{2\top}) \\ b_\top(x_2) & \text{if} \quad x_2 \in [x_{2\top}, 1]. \end{cases} \qquad (7)$$

Moreover, $\zeta_1(0) = \zeta_1(1)$, $\zeta_1(0) \leq \zeta_1(x_2)$ for all $x_2 \in [0,1]$. Similarly, $\rho_2^*(x) = \mathbb{1}\{x_2 > \zeta_2(x_1)\}$ where

$$\zeta_2(x_1) = \begin{cases} b_\perp'(x_1) & \text{if} \quad x_1 \in [0, x_{1\perp}] \\ b_\equiv(x_1) & \text{if} \quad x_1 \in (x_{1\perp}, x_{1\top}) \\ b_\top'(x_1) & \text{if} \quad x_1 \in [x_{1\top}, 1] \end{cases} \qquad (8)$$

such that $\zeta_2(0) = \zeta_2(1)$, $\zeta_2(0) \leq \zeta_2(x_2)$ for all $x_1 \in [0,1]$.

As can be seen from Figure 1(b), our characterization in Proposition 4.5 matches with the empirical observation obtained by solving the LP in (5).

The assumptions that $b_\equiv(x_2)$ is decreasing in $x_2$, and $b_\equiv(x_2)$ and $b_\top(x_2)$ have a unique intercept are hard to prove due to the complicated nature of the gaps $\kappa_i(x) - \kappa_j(x)$ between actions 4-1 and 4-3 in terms of $x$. This stems from the fact that diagnosis depends on the posterior, not just the likelihood. We provide empirical evidence for these assumptions. We show monotonicity of $b_\equiv(x_2)$ and unique intersection of $b_\equiv(x_2)$ and $b_\top(x_2)$ across a wide range of parameters in Appendix C. If these assumptions are violated, the decision boundaries may not be as simple as the ones stated in Proposition 4.5. However, this does not diminish the practical applicability of our proposed joint screening program. Another intriguing task is to understand the unified effect of the risk vector on the optimal referral decision. To this end, we also characterize the optimal referral decision when the screening activation of each disease is restricted to depend only on the risk of that disease.

**Proposition 4.6.** Restricted to the set of independent referral decisions $\rho_n(x) = \bar{\rho}_n(x_n)$, which depend solely on the variables related to disease $n$ when deciding whether to screen, the optimal referral decision has the form $\bar{\rho}_1^*(x_1) = \mathbb{1}\{x_1 > \bar{\zeta}_1\}$ and $\bar{\rho}_2^*(x_2) = \mathbb{1}\{x_2 > \bar{\zeta}_2\}$ for some thresholds $\bar{\zeta}_1, \bar{\zeta}_2 \in [0,1]$. When the budget allocated for screening of each disease under independent referral decision matches the budget allocations in the optimal unified referral decision, the thresholds satisfy $\zeta_1(0) = \min_{x_2 \in [0,1]} \zeta_1(x_2) \leq \bar{\zeta}_1 \leq \max_{x_2 \in [0,1]} \zeta_1(x_2)$ and $\zeta_n(0) \leq \bar{\zeta}_n$.

As can be seen from Figure 1(a), our characterization in Proposition 4.6 matches with the empirical observation for the optimal independent referral decision. The differences between the actions induced by the optimal referral decisions in Propositions 4.5 and 4.6 are illustrated in Figure 1(c). If the prior probability of having the other disease is zero (e.g., $x_2 = 0$ in the case of $N = 2$), then the activation threshold of the current disease in the unified case is less than the value in independent case ($\zeta_n(0) \leq \bar{\zeta}_n$).

## 5. In-Silico Experiments

### 5.1. Setup

We consider $N = 2$ diseases and assume that the earliest start time of screening is 50 years of age which is in line with the most common recommended age for screening programs in the US, such as for prostate, breast and colorectal cancer (Smith et al., 2001; Zoorob et al., 2001). As another example, Ma et al. (2023) calculate life expectancy starting at age 50 for cardiovascular disease. For each iteration, survival time $T_0$ (in years) is given as 40, where we round the value of 36.5-year average life expectancy of people with high cardiovascular health (CVH) score starting from age 50, as determined by (Ma et al., 2023). Adverse event times $T_n$, $n \in [N]$ (in years) are generated from $\mathcal{N}(\mu_n, \sigma^2)$. In (Ma et al., 2023), the average life expectancy with low and moderate CVH is estimated as 27.3 and 32.9 years respectively, hence we take the average of these two as $\mu_n = 30.1$ for the mean adverse event times, and we set $\sigma^2 = 1$.[6]

We set $p_X(x) \propto f_{\text{Beta}}(x_1; \alpha, \beta) f_{\text{Beta}}(x_2; \alpha, \beta)$ where $f_{\text{Beta}}(x; \alpha, \beta)$ is the pdf of the Beta distribution. Here, we assume that $X_1$ and $X_2$ are independent, however, they are distributed with the same coefficients. When experimenting with other joint distributions where $X_1$ and $X_2$ are not necessarily independent, we obtain nearly identical results with similar boundary shapes. This distribution is chosen for ease of implementation and to see the effects of varying $\alpha$ and $\beta$. The samples $Y_n(t)$ are generated by adding Gaussian noise to true measurement $s_n(x, t)$ which is a uniform distribution between 0 and 1, given that $\Theta_n = 1$, and if $\Theta_n = 0$, we set $s_n(x, t) = 0$. The noise is sampled from a normal distribution with a standard deviation of $\sigma = 0.5$, such that $Y_n(x, t) = s_n(x, t) + e_n(x, t)$, where $s_n(x, t) \sim \text{Uniform}(0, 1)$ and $e_n(t) \sim \mathcal{N}(0, \sigma^2)$. We sample once and use $s_n(t) = s_n$ for all $t$, but the error $e_n(x, t)$ is sampled independently for each $t$, leading to $Y(t) \perp\!\!\!\perp Y(t') \mid \Theta$ for $t' \neq t$.

We assume that the screening policy for each disease when active screens periodically every 1 year, until an adverse event happens or a positive diagnosis is made. We take the

---
[6]Code is available at https://github.com/ynarter/UniScreen.

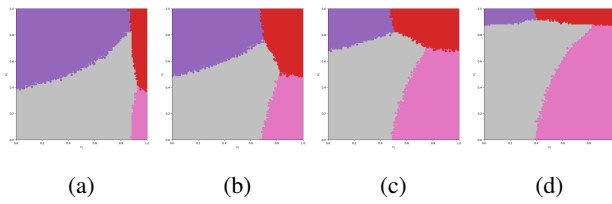

(a)      (b)      (c)      (d)

Figure 4: Varying the screening costs ($c_n$) for the diseases with other parameters fixed, (a) $c_1 = 6, c_2 = 1$, (b) $c_1 = 2, c_2 = 1$, (c) $c_1 = 1, c_2 = 2$, (d) $c_1 = 1, c_2 = 6$.

default budget $B = 10$, $\alpha = \beta = 5$ and the individual screening costs as $c_1 = c_2 = 1$.

At each Monte Carlo iteration, we simultaneously run $M = 200$ independent simulations for each of the $N = 10,000$ patient feature vectors $x = (x_1, x_2)$ to estimate the expected outcomes under various screening actions. Specifically, for each $x$ and action $a$, we compute empirical expectations of the form $\mathbb{E}[h(T^*) \mid a, x] \approx \frac{1}{M} \sum_{i=1}^{M} h(T_i^*(x, a))$, where $h(T^*) \in \{\min\{T_0, T^*\}, \#_n(\min\{T_0, T^*\})\}$, and $T_i^*(x, a)$ denotes the $i$th simulated outcome under action $a$. We fix the diagnosis threshold to $\gamma_n = 0.95$ throughout.

### 5.2. Results

Optimal actions/referral decisions are presented in Figure 1(b) (refer to Figure 11 for a detailed view). In the plot, the characterizations can be observed, and therefore be verified. We observe a symmetry for the activation boundaries of the first and second disease, the non-increasing or non-decreasing properties of the sub-referral decisions as derived, also that $\zeta_1(0) = \zeta_1(1)$.

A series of experiments was conducted with systematic variations in key parameters $B, c_n, \mu_n = \mathbb{E}[T_n], \alpha, \beta$, as presented in Appendix D.1. As an example, we observe that when we vary the individual screening costs ($c_n$) for diseases, an increase in the cost of screening for a particular disease leads the model to show a preference for referral decisions targeting only the alternative, lower-cost disease (see Figure 4). This behavior highlights the model's sensitivity to cost constraints, as it prioritizes affordable screening options under limited resources.

Next, we examined the values of $\kappa_i$ across the four policies, selecting the policy with the highest $\kappa_i$ at each point. We generated decision boundaries by selecting policies with the largest $\kappa$ values, and recreated the decision points found in our mathematical proof by locating intersections where one $\kappa_i$ surpasses the others (see Figure 16b). A comparison of these boundaries, derived by selecting the maximum $\kappa_i$ (per our theoretical framework) and those from LP solution, confirms that the boundary shapes are nearly identical (see

Figure 17).

To find the optimal independent referral decision, we modified our framework to separately optimize the referral policies for the two diseases. The budget is distributed equally to the two diseases such that $B_1 = B_2 = 5$, while the budget for our model is $B = 10$. The independent policies are then combined into a single referral decision using bitwise operations. In contrast, our model determines a joint referral decision by solving a LP that simultaneously considers both diseases. The resulting plot can be seen in Figure 1(a). As expected, the thresholds are constant as we had shown in the proof, and they lie between the minimum and maximum value of the unified screening thresholds, which leads to both gray (no screening) and colored (screening) points in the policy difference plot (Figure 1(c)).

We determine the survival times $r_{a,x}$ for both models (see Appendix D.3 for details) to compare differences in survival outcomes (Figure 5). We also plot the average survival times for each $x_1$ over all $x_2$'s for better visualization, and determine the overall mean of the survival times for both models, as displayed in Figure 2. We observe that the overall mean for the unified model is 37.70, whereas for the independent model, this is 37.47, which indicates an overall better performance. Moreover, as detailed in Appendix E, we verify the statistical significance of our results through further Monte Carlo simulations, demonstrating that the observed survival benefits of the unified approach are robust and do not result from random variation.

As observed in Figures 2 and 5, our model performs significantly better than independent screening at points where the risk of one disease outweighs the other, particularly around the optimal activation threshold of the disease in the independent case. This demonstrates the effectiveness of prioritizing screenings based on individual risks while allowing for joint screenings when appropriate, as previously discussed in the introduction. In the middle points of the grid where the two risks are moderate and close to each other, independent screening performs better. This is caused by the emphasis we put on the relatively high risks of diseases, where we allow for screenings as opposed to independent screening (see Figure 1(c)). Joint screenings ($a = 4$) also occur more often in the unified model (2363 vs. 1847), suggesting more efficient scheduling and improved survival for high-risk patients.

Our results display the limitations of independent policies in addressing overlaps and interactions between diseases. Independent screening programs often miss opportunities to improve patient care by failing to consider how screening for one disease can indirectly benefit another. For example, treating one condition, such as heart disease, can enhance the effectiveness of screening for another, such as lung cancer. The unified model overcomes these limitations by

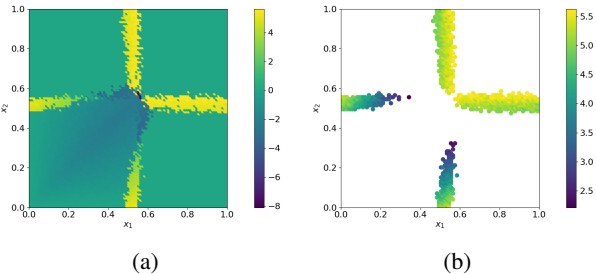

(a)                         (b)

Figure 5: (a) Difference of survival times ($\mathbb{E}\left[\min\{T_0, T^*\}|a, x\right]$) between unified and independent screening (unified $-$ independent) (b) points at which the survival time for unified screening exceeds independent screening.

considering these interactions, leading to better resource allocation and survival/health outcomes.

## 6. Conclusion, Limitations, Future Research

We proposed a unified screening model that accounts for the interactions between multiple diseases, to address limitations of traditional independent screening programs. By incorporating multiple disease risks into a single optimization framework, our approach enables efficient resource allocation under budget limits. Our experimental results demonstrate that the mathematical characterizations of optimal policies align with the numerical solution of the convex program. Additionally, the unified model increases the number of joint screenings, improving scheduling efficiency and enhancing survival outcomes, particularly for patients with higher relative risks for one disease.

A limitation of our work is that it provided the theoretical characterization of the optimal referral decision under $N = 2$ and uniform risk profile. Nevertheless, we studied the effect of non-uniform risk profiles in the experiments. Our work underscores the importance of considering competing risks and disease interactions in screening policy design. For future work, we aim to develop a detailed roadmap for implementing our model in clinical practice.

Our claims are based on the premise that screening is beneficial, and should be adapted when there are enough resources. In our case, the only detrimental effect of screening is a false positive, which can be controlled by adjusting the threshold for the likelihood ratio test. We agree that screening does not always offer benefits and sometimes can even be harmful, as in the case of overdiagnosis. A recent review paper for cancer screening (Bretthauer et al., 2023) argues that many common cancer screening programs do not significantly prolong life. Investigating optimal resource allocation by considering detrimental effects of screening is left as future research.

## Acknowledgments

The work of Cem Tekin was supported by Turkish National Academy of Sciences Distinguished Young Scientist Award Program TÜBA-GEBİP-2023, and the Scientific and Technological Research Council of Türkiye (TÜBİTAK) 2024 Incentive Award.

## Impact Statement

This work aims to advance the field of optimization by addressing the problem of unified screening for multiple diseases. By developing an optimization model that integrates competing risks and proposes an optimal referral decision, the research has the potential to improve healthcare decision-making and resource allocation. Once an implementation of these methods in clinical practice is achieved, healthcare systems could significantly improve early disease detection and treatment allocation, which can ultimately lead to better patient outcomes and reduced healthcare costs. The societal benefits of this work include enhancing public health outcomes by enabling more accurate and efficient screening strategies, particularly in resource-constrained settings. Moreover, the methodology could contribute to reducing health disparities by tailoring screening programs to patient-specific characteristics.

Ethical considerations include ensuring fairness and equity in the application of the proposed screening policies. Care must be taken to prevent unintended biases, particularly when implementing these models in diverse populations. The algorithms rely on patient data, emphasizing the need for secure data handling and adherence to privacy regulations to safeguard sensitive information. While the focus of this work is primarily methodological, the potential societal consequences highlight the importance of interdisciplinary collaboration with healthcare professionals and policymakers to ensure ethical and equitable implementation in real-world settings.

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

# A. Additional Discussion on Related Work

Prior research has predominantly focused on optimizing sampling schedules or decision-making processes for single-disease screenings. These strategies often involve fixed thresholds like age and weight (Association, 2018; Handelsman, 2015), to determine or evaluate inclusion criteria (Tomaszewski et al., 2022; Bauer et al., 2015; Fernandez et al., 1991; Yu et al., 2021; Xiao et al., 2022; Idrees et al., 2021; Wang et al., 2024b) and screening policy frequency (Hemmati et al., 2024; Wu & Suen, 2022; Wang et al., 2024b; Wu et al., 2024). Most of the existing studies focus on Markov models and processes (Wu et al., 2024; Wu & Suen, 2022; Alagoz, 2011; Ayer et al., 2012; Zhang et al., 2012a;b; Maillart et al., 2008; Kamalzadeh et al., 2021; Steimle & Denton, 2017). For example, Maillart et al. (2008) evaluate a partially observed Markov chain model to balance breast cancer mortality risk and mammogram frequency.

Cost-effectiveness analyses have also been a critical component of the literature (Toumazis et al., 2021; Fritzen et al., 2018; Cusick et al., 2023; Meng et al., 2024; Sharma et al., 2022; Areia et al., 2022; Huang et al., 2022), particularly for identifying the optimal trade-off between screening costs and health benefits (Xia et al., 2024; Qin et al., 2022; 2024; Wang et al., 2024a; Bao et al., 2022). Resource constraints, such as limited budgets, personnel, and equipment (Yadla et al., 2024; Khoza-Shangase et al., 2017), play a fundamental role in these analyses, as they directly impact the feasibility and scalability of screening programs (Cevik et al., 2018; Bansal et al., 2020). For instance, constrained budgets may require prioritizing patients or conditions (Cevik et al., 2018), while limited personnel and equipment can restrict the number of screenings performed in a given timeframe. Teh et al. (2015) analyze opportunistic mammography screening in resource-constrained healthcare systems. They demonstrate that their approach is effective, particularly for high-risk individuals, and recommend population-based screening for women aged 50 and above if resources permit. Bansal et al. (2020) address this by applying a Markov decision process model to determine optimal breast cancer screening schedules under varying infrastructure capacities. However, to the best of our knowledge, none of the existing methods address both resource constraints and competing risks of multiple diseases simultaneously, which leaves a gap in optimizing multi-disease screening programs under such conditions.

A competing risks problem in our approach refers to the assessment of the likelihood of health benefits from an intervention, considering the presence of other potential outcomes that may interfere with or prevent the occurrence of the event of interest (Varadhan et al., 2010). This approach enables the analysis of both the timing of the first observed event and its type (Wolbers et al., 2014). Competing risk analyses have been tackled in the literature (Allignol et al., 2011; Wolbers et al., 2014; Satagopan et al., 2004; Pintilie, 2011; Cho et al., 2022), which highlights its importance.

Several studies highlight the benefits of joint screening programs (Fan et al., 2024; Huang et al., 2023). Liang et al. (2016) assess joint detection using 3D-CPA, HR-HPV, and TCT, which improves accuracy and sensitivity in cervical disease screening. Regarding the screening for multiple diseases, Wang et al. (2022) propose a novel multi-view attention network for screening multiple retinal diseases from optical coherence tomography (OCT) images. Alternating from the "single disease" approach, Hajjar & Alagoz (2023) develop a stochastic modeling framework that personalizes screening decisions for an index disease by accounting for the dynamic and stochastic nature of chronic conditions. Moreover, for the detection of multiple diseases, frameworks leveraging deep learning methods exist (Ampavathi & Saradhi, 2021; George et al., 2024; Satheeskumaran et al., 2024; Arumugam et al., 2023; Men et al., 2021; Khadir et al., 2023; Sudha et al., 2024).

Our work on designing disease screening programs, which is about making decisions based on accumulating evidence to optimize outcomes under uncertainty, is closely related to sequential hypothesis testing (Chernoff, 1992; Alaa & Van Der Schaar, 2016; Drugowitsch et al., 2014; 2012; Shvartsman et al., 2015; Shiryaev, 2007; Peskir & Shiryaev, 2006). With multiple diseases, the general problem is an instance of sequential hypothesis testing with multiple hypotheses. Prior work, such as by Alaa & Van Der Schaar (2016) examines resource allocation with a Bayesian model in single-hypothesis (disease) scenarios but does not extend to the complexities of multi-disease contexts. However, in our context, multiple diseases correspond to multiple hypotheses, which transforms our problem into an active learning challenge, where actions influence the acquisition of information across diseases.

# B. Proofs of the Statements in Section 4

Recall that when $x_n \geq \gamma_n$, disease $n$ is immediately diagnosed (no need for screening). A similar case happens for $x_n = 0$ (no screening necessary). Hence, in practice, whether to screen or not is only a question when $x_n \in (0, \gamma_n)$. Nevertheless, to characterize differences between the benefits of different screening actions, we will compare their rewards and costs for all $x_n \in [0, 1]$. In comparing the effectiveness of different screening policies, we assume that if a patient is admitted for

screening, the screening policy collects at least one screening sample before the diagnosis.

## B.1. Technical Results Related to Forms of $r_{a,x}$ and $m_{a,x}$

Let $\mathcal{U}_n := \{\Theta_n = 1\} \wedge \{t_n^* \geq T_n\}$ represent the event that disease $n$ is present and its adverse event occurs before it is correctly diagnosed. Let $S_{T^*}(t, x, a) := \mathbb{P}\{T^* > t | a, x\}$. We first provide a decomposition lemma for $S_{T^*}(t, x, a)$.

**Lemma B.1.** *For any* $t \in \mathbb{N}_+$, $x \in [0,1]^N$, $a \in \mathcal{A}$,

$$S_{T^*}(t, x, a) = \prod_{n \in [N]} \left( \underbrace{\mathbb{P}\{T_n > t | a, x\}}_{\substack{\text{no adverse event occurs} \\ \text{until time } t}} + \underbrace{\mathbb{P}\{T_n \leq t, \mathcal{O}_n | a, x\}}_{\substack{\text{an adverse event has occurred} \\ \text{but is diagnosed on time}}} \right). \tag{9}$$

*Proof.*

$$\begin{aligned}
S_{T^*}(t, x, a) &= \mathbb{P}\{T^* > t | a, x\} \\
&= \mathbb{P}\{\min\{T_n : n \in [N], \mathbb{1}(\mathcal{U}_n) = 1\} > t | a, x\} \\
&= \mathbb{P}\{\min\{T_n : n \in [N], \{\mathbb{1}(\mathcal{U}_n) = 1\} \vee \{\Theta_n = 0\}\} > t | a, x\} \tag{10} \\
&= \mathbb{P}\{\min\{T_n : n \in [N], \mathbb{1}(\mathcal{O}_n) = 0\} > t | a, x\} \\
&= \mathbb{P}\{\cap_{n \in [N] : \mathbb{1}(\mathcal{O}_n)=0}\{T_n > t\} | a, x\} \\
&= \mathbb{P}\{\cap_{n \in [N]}\{T_n > t - \mathbb{1}(\mathcal{O}_n)t\} | a, x\} \tag{11} \\
&= \prod_{n \in [N]} \mathbb{P}\{T_n > t - \mathbb{1}(\mathcal{O}_n)t | a, x\} \tag{12} \\
&= \prod_{n \in [N]} (1 - \mathbb{P}\{T_n \leq t - \mathbb{1}(\mathcal{O}_n)t | a, x\}) \\
&= \prod_{n \in [N]} (1 - \mathbb{P}\{T_n \leq t - \mathbb{1}(\mathcal{O}_n)t, \mathcal{O}_n | a, x\} - \mathbb{P}\{T_n \leq t - \mathbb{1}(\mathcal{O}_n)t, \neg\mathcal{O}_n | a, x\}) \\
&= \prod_{n \in [N]} (1 - \mathbb{P}\{T_n \leq 0, \mathcal{O}_n | a, x\} - \mathbb{P}\{T_n \leq t, \neg\mathcal{O}_n | a, x\}) \\
&= \prod_{n \in [N]} (1 - \mathbb{P}\{T_n \leq t, \neg\mathcal{O}_n | a, x\}) \tag{13} \\
&= \prod_{n \in [N]} \left( \mathbb{P}\{T_n > t | a, x\} + \mathbb{P}\{T_n \leq t, \mathcal{O}_n | a, x\} \right) \tag{14}
\end{aligned}$$

where (10) is due to $T_n = \infty$ iff $\Theta_n = 0$, (12) holds since the random variables $T_n + \mathbb{1}(\mathcal{O}_n)t$ are conditionally independent of each other given $a, x$, (11) and (13) is due to the fact that $T_n$ is a positive random variable, lastly, (14) holds given the partition $\mathbb{P}\{T_n \leq t, \neg\mathcal{O}_n | a, x\} + \mathbb{P}\{T_n \leq t, \mathcal{O}_n | a, x\} + \mathbb{P}\{T_n > t | a, x\} = 1$. Note that if a screening is not performed for a disease with index $n$ based on the given action $a$, the term $\mathbb{P}\{T_n \leq t, \mathcal{O}_n | a, x\}$ is simply zero since there will be no diagnosis, and $T^*$ will simply denote the first adverse event that occurs ($T_n$).

$\square$

Next, we make some assumptions under which strict monotonicity of functions $\mathbb{P}\{T_n > t | a, x\}$ and $\mathbb{P}\{T_n \leq t, \mathcal{O}_n | a, x\}$ in $x$ hold.

**Assumption B.2.** (i). The support of $T_n$ is $[T_0]$, for all $n \in [N]$; (ii) When action $a$ screens for disease $n$, for all $x_n \in (0, 1]$, $\mathbb{P}\{t_n^* < t, T_n = t | a, x_n, \Theta_n = 1\} > 0$.

The next lemma characterizes how $\mathbb{P}\{T_n > t | a, x\}$ in (9) and $\mathbb{P}\{T_n \leq t | a, x\}$ change with $x$.

**Lemma B.3.** *Let* $F_{T_n}(t) := F_{T_n | \Theta_n}(t | 1)$. *For any* $t \in \mathbb{N}_+$, $x \in [0,1]^N$, $a \in \mathcal{A}$, $\mathbb{P}\{T_n > t | a, x\} = 1 - F_{T_n}(t)x_n$ is *non-increasing in* $x_n$, $\mathbb{P}\{T_n \leq t | a, x\} = F_{T_n}(t)x_n$ is *non-decreasing in* $x_n$. *Moreover, if the support of* $T_n$ *is* $[T_0]$, *then* $\mathbb{P}\{T_n > t | a, x\}$ *is strictly decreasing in* $x_n$, *and* $\mathbb{P}\{T_n \leq t | a, x\}$ *is strictly increasing in* $x_n$ *for* $t \in [T_0 - 1]$. *Both functions are constant with respect to* $x_m$, $m \neq n$.

*Proof.* Since $\mathbb{P}\{T_n > t|a,x\}$ is independent of the screening action $a$, we can drop the conditioning on $a$. Also, if the disease $n$ is not present, i.e., $\Theta_n = 0$, then $T_n = \infty$. Using this, we obtain:

$$
\begin{aligned}
\mathbb{P}\{T_n > t|a,x\} = \mathbb{P}\{T_n > t|x\} &= \mathbb{P}\{T_n > t, \Theta_n = 0|x\} + \mathbb{P}\{T_n > t, \Theta_n = 1|x\} \\
&= \mathbb{P}\{\Theta_n = 0|x\}\mathbb{P}\{T_n > t|\Theta_n = 0, x\} + \mathbb{P}\{\Theta_n = 1|x\}\mathbb{P}\{T_n > t|\Theta_n = 1, x\} \\
&= (1 - x_n) + x_n \mathbb{P}\{T_n > t|\Theta_n = 1, x\} \\
&= (1 - x_n) + x_n(1 - F_{T_n|\Theta_n}(t|1)) \\
&= 1 - F_{T_n}(t)x_n.
\end{aligned}
\tag{15}
$$

Since the CDF $F_{T_n}(t) \geq 0, \forall t$, we conclude that $\mathbb{P}\{T_n > t|x\}$ is non-increasing in $x_n$.

We also have

$$
\mathbb{P}\{T_n \leq t|a,x\} = 1 - \mathbb{P}\{T_n > t|a,x\} = F_{T_n}(t)x_n,
$$

which is non-decreasing in $x_n$. $\qquad\square$

Let $\mathcal{S}_n := \{\tau_{n1}, \ldots, \tau_{nS_n}\}$ represent the set of all screening times for disease $n$. Let $\mathcal{S}_n(i) := \{\tau_{nj} \in \mathcal{S}_n : \tau_{nj} < i\}$ represent the set of screening times of disease $n$ prior to time step $i$. The next lemma characterizes how $\mathbb{P}\{T_n \leq t, \mathcal{O}_n|a,x\}$ in (9) changes with $x$.

**Lemma B.4.** *For any $a \in \mathcal{A}$ such that $a$ does not screen for disease $n$, $\mathbb{P}\{T_n \leq t, \mathcal{O}_n|a,x\} = 0$ for all $t \in \mathbb{N}_+$ and $x \in [0,1]^N$. For any $a \in \mathcal{A}$ such that $a$ screens for disease $n$, for all $t \in \mathbb{N}_+$, $\mathbb{P}\{T_n \leq t, \mathcal{O}_n|a,x\}$ is (i) non-decreasing in $x_n$, (ii) strictly increasing in $x_n$ if there exists $i \in [t]$ such that $\mathbb{P}\{t_n^* < i, T_n = i|a, x, \Theta_n = 1\} > 0$, (iii) constant with respect to $x_m$, $m \neq n$, and (iv) $0$ when $x_n = 0$ and $F_{T_n}(t)$ when $x_n = 1$.*

*Proof.* First, consider the case when action $a$ does not screen for disease $n$. Note that a referral policy never makes a diagnosis if screening is not active for disease $n$. Hence, $\mathbb{P}\{T_n \leq t, \mathcal{O}_n|a,x\} = 0$ for all $x$, since disease $n$ will never be diagnosed on time by the referral policy if it is present. Next, consider the case when action $a$ screens for disease $n$. Observe that

$$
\begin{aligned}
\mathbb{P}\{T_n \leq t, \mathcal{O}_n|a,x\} &= \mathbb{P}\{T_n \leq t, t_n^* < T_n, \Theta_n = 1|a,x\} \\
&= \mathbb{P}\{T_n \leq t, t_n^* < T_n|a, x, \Theta_n = 1\}x_n \\
&= \sum_{i=1}^{\infty} \mathbb{P}\{T_n \leq t, t_n^* < T_n, T_n = i|a, x, \Theta_n = 1\}x_n \\
&= \sum_{i=1}^{t} \mathbb{P}\{T_n \leq t, t_n^* < T_n, T_n = i|a, x, \Theta_n = 1\}x_n \\
&= \sum_{i=1}^{t} \mathbb{P}\{t_n^* < i, T_n = i|a, x, \Theta_n = 1\}x_n.
\end{aligned}
\tag{16}
$$

Next, we will investigate how $\mathbb{P}\{t_n^* < i, T_n = i|a, x, \Theta_n = 1\}$ in (16) varies with $x$. Note that

$$
\begin{aligned}
\mathbb{P}\{t_n^* < i, T_n = i|a, x, \Theta_n = 1\} &= \mathbb{P}\{t_n^* < i|T_n = i, a, x, \Theta_n = 1\}\mathbb{P}\{T_n = i|a, x, \Theta_n = 1\} \\
&= \mathbb{P}\{t_n^* < i|T_n = i, a, x, \Theta_n = 1\}\mathbb{P}\{T_n = i|\Theta_n = 1\}.
\end{aligned}
\tag{17}
$$

Next, we further investigate $\mathbb{P}\{t_n^* < i|T_n = i, a, x, \Theta_n = 1\}$ in (17).

$$
\begin{aligned}
&\mathbb{P}\{t_n^* < i|T_n = i, a, X = x, \Theta_n = 1\} \\
&= \mathbb{P}\{\min\{t \in \mathcal{S}_n : \hat{\theta}_n(x,t) = 1\} < i|T_n = i, a, X = x, \Theta_n = 1\} \\
&= \mathbb{P}\{\exists t \in \mathcal{S}_n(i) : \hat{\theta}_n(x,t) = 1|T_n = i, a, X = x, \Theta_n = 1\} \\
&= \mathbb{P}\left\{\exists t \in \mathcal{S}_n(i) : \frac{X_n}{1 - X_n}\text{LR}_n(t) \geq \frac{\gamma_n}{1 - \gamma_n}\Big|T_n = i, a, X = x, \Theta_n = 1\right\}
\end{aligned}
$$

$$= \mathbb{P}\left\{\exists t \in \mathcal{S}_n(i) : \frac{x_n}{1-x_n}\mathrm{LR}_n(t) \geq \frac{\gamma_n}{1-\gamma_n}\Big| T_n = i, \Theta_n = 1\right\} \tag{18}$$

$$= \mathbb{P}\left\{\exists t \in \mathcal{S}_n(i) : \mathrm{LR}_n(t) \geq \frac{(1-x_n)\gamma_n}{x_n(1-\gamma_n)}\Big| T_n = i, \Theta_n = 1\right\}$$

$$= \mathbb{P}\left\{\max_{t \in \mathcal{S}_n(i)} \mathrm{LR}_n(t) \geq \frac{(1-x_n)\gamma_n}{x_n(1-\gamma_n)}\Big| T_n = i, \Theta_n = 1\right\}, \tag{19}$$

where (18) holds due to observations $Y_n(\tau_{ni})$, $i \in \mathcal{S}_n(i)$ being independent of $X$ given $\Theta_n$ and $T_n$. From (19), we conclude that $\mathbb{P}\{t_n^* < i | T_n = i, a, X = x, \Theta_n = 1\}$ is non-decreasing in $x_n$. By (17), this property also applies to $\mathbb{P}\{t_n^* < i, T_n = i | a, x, \Theta_n = 1\}$. By (17), $\mathbb{P}\{T_n \leq t, \mathcal{O}_n | a, x\}$ is sum of non-negative non-decreasing functions multiplied by $x_n$, so it is non-decreasing in $x_n$. Moreover, it is increasing in $x_n$ if at least one of these functions is positive. Note that none of the terms above depend on $x_m$, $m \neq n$, so $\mathbb{P}\{T_n \leq t, \mathcal{O}_n | a, x\}$ does not vary with $m$. When $x_n = 0$, $\mathbb{P}\{T_n \leq t, \mathcal{O}_n | a, x\} = 0$ by (16). When $x_n = 1$,

$$\sum_{i=1}^{t} \mathbb{P}\{t_n^* < i, T_n = i | a, x, \Theta_n = 1\} x_n = \sum_{i=1}^{t} \mathbb{P}\{T_n = i | a, x, \Theta_n = 1\} x_n$$

$$= \sum_{i=1}^{t} \mathbb{P}\{T_n = i | \Theta_n = 1\} x_n = \mathbb{P}\{T_n \leq t | \Theta_n = 1\} x_n = F_{T_n}(t).$$

$\square$

Using Lemmas (B.1), (B.3) and (B.4), $S_{T^*}(t, x, a)$, $a \in [4]$ can be written as

$$S_{T^*}(t, x, 1) = \mathbb{P}\{T_1 > t | x_1\}\mathbb{P}\{T_2 > t | x_2\}, \tag{20}$$

$$S_{T^*}(t, x, 2) = (\mathbb{P}\{T_1 > t | x_1\} + \mathbb{P}\{T_1 \leq t, \mathcal{O}_1 | 2, x_1\})\mathbb{P}\{T_2 > t | x_2\}, \tag{21}$$

$$S_{T^*}(t, x, 3) = \mathbb{P}\{T_1 > t | x_1\}(\mathbb{P}\{T_2 > t | x_2\} + \mathbb{P}\{T_2 \leq t, \mathcal{O}_2 | 3, x_2\}), \tag{22}$$

$$S_{T^*}(t, x, 4) = (\mathbb{P}\{T_1 > t | x_1\} + \mathbb{P}\{T_1 \leq t, \mathcal{O}_1 | 2, x_1\})(\mathbb{P}\{T_2 > t | x_2\} + \mathbb{P}\{T_2 \leq t, \mathcal{O}_2 | 3, x_2\}). \tag{23}$$

We define the following gaps:

$$\Delta_{2,1}^T(t, x) := S_{T^*}(t, x, 2) - S_{T^*}(t, x, 1) = \mathbb{P}\{T_1 \leq t, \mathcal{O}_1 | 2, x_1\}\mathbb{P}\{T_2 > t | x_2\}. \tag{24}$$

$\Delta_{2,1}^T(t, x)$ is non-decreasing in $x_1$ and non-increasing in $x_2$. Under Assumption B.2, it is strictly increasing in $x_1$ and strictly decreasing in $x_2$. Moreover, $\Delta_{2,1}^T(t, x_1 = 0, x_2) = 0$.

$$\Delta_{3,1}^T(t, x) := S_{T^*}(t, x, 3) - S_{T^*}(t, x, 1) = \mathbb{P}\{T_2 \leq t, \mathcal{O}_2 | 3, x_2\}\mathbb{P}\{T_1 > t | x_1\}. \tag{25}$$

$\Delta_{3,1}^T(t, x)$ is non-increasing in $x_1$ and non-decreasing in $x_2$. Under Assumption B.2, it is strictly decreasing in $x_1$ and strictly increasing in $x_2$. Moreover, $\Delta_{3,1}^T(t, x_1, x_2 = 0) = 0$.

$$\Delta_{4,2}^T(t, x) := S_{T^*}(t, x, 4) - S_{T^*}(t, x, 2) = (\mathbb{P}\{T_1 > t | x_1\} + \mathbb{P}\{T_1 \leq t, \mathcal{O}_1 | 2, x_1\})\mathbb{P}\{T_2 \leq t, \mathcal{O}_2 | 3, x_2\}. \tag{26}$$

$\Delta_{4,2}^T(t, x)$ is non-decreasing in $x_2$. Under Assumption B.2, it is strictly increasing in $x_2$. Moreover, $\Delta_{4,2}^T(t, x_1, x_2 = 0) = 0$.

$$\Delta_{4,3}^T(t, x) := S_{T^*}(t, x, 4) - S_{T^*}(t, x, 3) = \mathbb{P}\{T_1 \leq t, \mathcal{O}_1 | 2, x_1\}(\mathbb{P}\{T_2 > t | x_2\} + \mathbb{P}\{T_2 \leq t, \mathcal{O}_2 | 3, x_2\}). \tag{27}$$

$\Delta_{4,3}^T(t, x)$ is non-decreasing in $x_1$. Under Assumption B.2, it is strictly increasing in $x_1$. Moreover, $\Delta_{4,3}^T(t, x_1 = 0, x_2) = 0$. Define the last gap as

$$\Delta_{4,1}^T(t, x) := (\mathbb{P}\{T_1 > t | x_1\} + \mathbb{P}\{T_1 \leq t, \mathcal{O}_1 | 2, x_1\})(\mathbb{P}\{T_2 > t | x_2\} + \mathbb{P}\{T_2 \leq t, \mathcal{O}_2 | 3, x_2\}) \tag{28}$$

$$- \mathbb{P}\{T_1 > t | x_1\}\mathbb{P}\{T_2 > t | x_2\}$$

$$= \mathbb{P}\{T_1 \leq t, \mathcal{O}_1 | 2, x_1\}(\mathbb{P}\{T_2 > t | x_2\} + \mathbb{P}\{T_2 \leq t, \mathcal{O}_2 | 3, x_2\}) + \mathbb{P}\{T_1 > t | x_1\}\mathbb{P}\{T_2 \leq t, \mathcal{O}_2 | 3, x_2\}$$

$$= \mathbb{P}\{T_2 \le t, \mathcal{O}_2|3, x_2\}(\mathbb{P}\{T_1 > t|x_1\} + \mathbb{P}\{T_1 \le t, \mathcal{O}_1|2, x_1\}) + \mathbb{P}\{T_2 > t|x_2\}\mathbb{P}\{T_1 \le t, \mathcal{O}_1|2, x_1\}. \quad (29)$$

The following display lists the relations between these gaps.

$$\Delta^T_{4,1}(t, x) \ge \Delta^T_{4,3}(t, x) \ge \Delta^T_{2,1}(t, x), \;\; \Delta^T_{4,1}(t, x) \ge \Delta^T_{4,2}(t, x) \ge \Delta^T_{3,1}(t, x).$$

As the next step, we will derive an expression for $\mathbb{E}[\min\{T_0, T^*\}|a, x]$. Since $\min\{T_0, T^*\}$ is non-negative, we write

$$\mathbb{E}[\min\{T_0, T^*\}|a, x] = \sum_{t=0}^{\infty} \mathbb{P}\{\min\{T_0, T^*\} > t|a, x\} = \sum_{t=0}^{\infty} \mathbb{P}\{T_0 > t, T^* > t|a, x\} = \sum_{t=0}^{T_0-1} \mathbb{P}\{T^* > t|a, x\}$$

$$= \sum_{t=0}^{T_0-1} S_{T^*}(t, x, a).$$

We define the survival gaps as $\Delta^T_{i,j}(x) := \mathbb{E}[\min\{T_0, T^*\}|i, x] - \mathbb{E}[\min\{T_0, T^*\}|j, x] = r_{i,x} - r_{j,x}$. Note that using the expression in the above display, these can be written in terms of gaps $\Delta^T_{i,j}(t, x)$ as $\Delta^T_{i,j}(x) = \sum_{t=0}^{T_0-1} \Delta^T_{i,j}(t, x)$. The characteristics of $\Delta^T_{i,j}(t, x)$s directly apply to $\Delta^T_{i,j}(x)$ due to summation preserving monotonicity.

**Lemma B.5.** *Under Assumption B.2: (i) $\Delta^T_{2,1}(x)$ is strictly increasing in $x_1$, strictly decreasing in $x_2$, and $\Delta^T_{2,1}(x_1 = 0, x_2) = 0$; (ii) $\Delta^T_{3,1}(x)$ is strictly decreasing in $x_1$, strictly increasing in $x_2$, and $\Delta^T_{3,1}(x_1, x_2 = 0) = 0$; (iii) $\Delta^T_{4,2}(x)$ is strictly increasing in $x_2$, and $\Delta^T_{4,2}(x_1, x_2 = 0) = 0$; (iv) $\Delta^T_{4,3}(x)$ is strictly increasing in $x_1$, and $\Delta^T_{4,3}(x_1 = 0, x_2) = 0$; (v) The following relations hold.*

$$\Delta^T_{4,1}(x) \ge \Delta^T_{4,3}(x) \ge \Delta^T_{2,1}(x), \;\; \Delta^T_{4,1}(x) \ge \Delta^T_{4,2}(x) \ge \Delta^T_{3,1}(x). \quad (30)$$

Next, we focus on the expected costs of the actions. Recall that $m_{a,x} = c_1\mathbb{E}[\#_1|a, x] + c_2\mathbb{E}[\#_2|a, x]$. Under (i) no screening ($a = 1$), $m_{1,x} = 0$; (ii) screening only for disease 1 ($a = 2$), $m_{2,x} = c_1\mathbb{E}[\#_1|a = 2, x]$; (iii) screening only for disease 2 ($a = 3$), $m_{3,x} = c_2\mathbb{E}[\#_2|a = 3, x]$, screening for both diseases $m_{4,x} = c_1\mathbb{E}[\#_1|a = 4, x] + c_2\mathbb{E}[\#_2|a = 4, x] = c_1\mathbb{E}[\#_1|a = 2, x] + c_2\mathbb{E}[\#_2|a = 3, x]$, for all $x \in [0, 1]^2$.

We analyze $\mathbb{E}[\#_n(\min\{T_0, T_n, t^*_n\})|a, x]$.

**Lemma B.6.** *When action $a$ screens for disease $n$, $\mathbb{E}[\#_n(\min\{T_0, T_n, t^*_n\})|a, x]$ is non-increasing in $x_n$ and constant in $x_m$, $m \ne n$. Otherwise, it is constant in $x_n$. $\mathbb{E}[\#_n(\min\{T_0, T_n, t^*_n\})|a, x_n = 1] = 1$, $\mathbb{E}[\#_n(\min\{T_0, T_n, t^*_n\})|a, x_n = 0] = S_n$.*

*Proof.* Let $S_{ni}$ represent the indicator variable that disease $n$ is screened for the $i$th time.

$$\mathbb{E}[\#_n(\min\{T_0, T_n, t^*_n\})|a, x] = \sum_{i=1}^{S_n(T_0)} \mathbb{P}\{S_{ni} = 1|a, x\} = \sum_{i=1}^{S_n(T_0)} \mathbb{P}\{T_n > \tau_{ni}, t^*_n > \tau_{ni}|a, x\}, \quad (31)$$

where

$$\mathbb{P}\{T_n > \tau_{ni}, t^*_n > \tau_{ni}|a, x\} = \mathbb{P}\{T_n > \tau_{ni}, t^*_n > \tau_{ni}, \Theta_n = 0|a, x\} + \mathbb{P}\{T_n > t, t^*_n > \tau_{ni}, \Theta_n = 1|a, x\}$$

$$= \mathbb{P}\{T_n > \tau_{ni}, t^*_n > \tau_{ni}|\Theta_n = 0, a, x\}(1 - x_n) + \mathbb{P}\{T_n > \tau_{ni}, t^*_n > \tau_{ni}|\Theta_n = 1, a, x\}x_n$$

$$= \mathbb{P}\{t^*_n > \tau_{ni}|\Theta_n = 0, a, x\}(1 - x_n) + \mathbb{P}\{T_n > \tau_{ni}, t^*_n > \tau_{ni}|\Theta_n = 1, a, x\}x_n$$

$$= \underbrace{\mathbb{P}\{t^*_n > \tau_{ni}|\Theta_n = 0, a, x_n\}}_{f(x_n)}(1 - x_n) + \underbrace{\mathbb{P}\{T_n > \tau_{ni}, t^*_n > \tau_{ni}|\Theta_n = 1, a, x_n\}}_{g(x_n)} x_n. \quad (32)$$

We have

$$\mathbb{P}\{t^*_n > \tau_{ni}|\Theta_n = 0, a, X_n = x_n\} = \mathbb{P}\{\min\{t \in \mathcal{S}_n : \hat{\theta}_n(x, t) = 1\} > \tau_{ni}|\Theta_n = 0, a, X_n = x_n\}$$

$$= \mathbb{P}\left\{\forall j \in [i] : \frac{X_n}{1 - X_n}\mathrm{LR}_n(\tau_{nj}) < \frac{\gamma_n}{1 - \gamma_n}\middle|\Theta_n = 0, X_n = x_n\right\}$$

$$= \mathbb{P}\left\{\max_{j\in[i]}\mathrm{LR}_n(\tau_{nj}) < \frac{(1-x_n)\gamma_n}{x_n(1-\gamma_n)}\bigg|\Theta_n = 0\right\},$$

which is non-increasing in $x_n$. Similarly,

$$\mathbb{P}\{t_n^* > \tau_{ni}, T_n > \tau_{ni}|\Theta_n = 1, a, X_n = x_n\}$$

$$= \sum_{i=1}^\infty \mathbb{P}\{t_n^* > \tau_{ni}, T_n > \tau_{ni}, T_n = i|\Theta_n = 1, a, X_n = x_n\}$$

$$= \sum_{i\in\mathbb{N}_+ : i > \tau_{ni}} \mathbb{P}\{t_n^* > \tau_{ni}, T_n > \tau_{ni}, T_n = i|\Theta_n = 1, a, X_n = x_n\}$$

$$= \sum_{i\in\mathbb{N}_+ : i > \tau_{ni}} \mathbb{P}\{t_n^* > \tau_{ni}, T_n = i|\Theta_n = 1, a, X_n = x_n\}$$

$$= \sum_{i\in\mathbb{N}_+ : i > \tau_{ni}} \mathbb{P}\{t_n^* > \tau_{ni}|T_n = i, \Theta_n = 1, a, X_n = x_n\}\mathbb{P}\{T_n = i|\Theta_n = 1\}$$

$$= \sum_{i\in\mathbb{N}_+ : i > \tau_{ni}} \mathbb{P}\left\{\max_{j\in[i]}\mathrm{LR}_n(\tau_{nj}) < \frac{(1-x_n)\gamma_n}{x_n(1-\gamma_n)}\bigg|T_n = i, \Theta_n = 1\right\}\mathbb{P}\{T_n = i|\Theta_n = 1\},$$

which is non-increasing in $x_n$. Also observe that

$$\mathbb{P}\left\{\max_{j\in[i]}\mathrm{LR}_n(\tau_{nj}) < \frac{(1-x_n)\gamma_n}{x_n(1-\gamma_n)}\bigg|\Theta_n = 0\right\} \geq \mathbb{P}\left\{\max_{j\in[i]}\mathrm{LR}_n(\tau_{nj}) < \frac{(1-x_n)\gamma_n}{x_n(1-\gamma_n)}\bigg|\Theta_n = 1\right\}.$$

Therefore, $\mathbb{P}\{t_n^* > \tau_{ni}|\Theta_n = 0, a, x_n\} \geq \mathbb{P}\{t_n^* > \tau_{ni}|\Theta_n = 1, a, x_n\} \geq \mathbb{P}\{t_n^* > \tau_{ni}, T_n > \tau_{ni}|\Theta_n = 1, a, x_n\}$.

By the above properties and the decomposition in (32), the following holds for $x_n < x_n'$:

$$f(x_n)(1-x_n) + g(x_n)x_n \geq f(x_n)(1-x_n') + g(x_n)x_n' \geq f(x_n')(1-x_n') + g(x_n')x_n',$$

by which we conclude that $\mathbb{P}\{T_n > \tau_{ni}, t_n^* > \tau_{ni}|a, x\}$ is non-increasing in $x_n$. Moreover, it does not depend on $x_m$, $m \neq n$. By (31), $\mathbb{E}[\#_n(\min\{T_0, T^*\})|a, x]$ is also non-increasing in $x_n$ (since it is the sum of non-increasing functions) and constant in $x_m$, $m \neq n$. $\square$

Let $\Delta_{i,j}^C(x) := m_{i,x} - m_{j,x}$ represent the expected cost gaps of the actions. The following lemma characterizes the dependence of these gaps to $x$. For the sake of simplicity of the analysis, assume that the costs of the first screening samples are excluded from the cost calculation. This makes sense when the goal is to manage the excess cost due to multiple screenings (when a patient is admitted to a screening program, usually many screenings take place over a long time horizon).

**Lemma B.7.** *(i)* $\Delta_{4,3}^C(x) = \Delta_{2,1}^C(x) = c_1(\mathbb{E}[\#_1(\min\{T_0, T_1, t_1^*\})|a, x] - 1)$ *is non-increasing in $x_1$ and constant in* $x_2$; *(ii)* $\Delta_{4,2}^C(x) = \Delta_{3,1}^C(x) = c_2(\mathbb{E}[\#_2(\min\{T_0, T_2, t_2^*\})|a, x] - 1)$ *is constant in $x_1$ and non-increasing in $x_2$; (iii)* $\Delta_{4,1}^C(x) = c_1(\mathbb{E}[\#_1(\min\{T_0, T_1, t_1^*\})|a, x] - 1) + c_2(\mathbb{E}[\#_2(\min\{T_0, T_2, t_2^*\})|a, x] - 1)$ *is non-increasing in $x_1$ and $x_2$.*

Let $\Delta_{i,j}(x) := \kappa_{i,x} - \kappa_{j,x}$. Then, $\Delta_{i,j}(x) = \Delta_{i,j}^T(x) - \lambda\Delta_{i,j}^C(x)$. We assume that the gap functions are continuous in $x_1$, $x_2$, which will hold under mild assumptions on the distributions of $Y_n$ and $T_n$, $n \in [2]$.

**Assumption B.8.** The functions $\Delta_{i,j}^T(x)$ and $\Delta_{i,j}^C(x)$, $i, j \in [4]$ are continuous in $x$. Therefore, $\Delta_{i,j}(x)$, $i, j \in [4]$ is continuous in $x$.

### B.2. Proof of Lemma 4.2

Consider $\rho_{2,1} = \mathbb{1}(\kappa_2(x) > \kappa_1(x))$. Under Assumption B.2, $\Delta_{2,1}(x)$ is strictly increasing in $x_1$ and strictly decreasing in $x_2$.

Fix $x_2 \in [0,1]$. Observe that $\Delta_{2,1}(x_1 = 0, x_2) = 0 - \lambda(S_1 - 1)c_1 < 0$ (screening programs are set to screen more than once) and $\Delta_{2,1}(x_1 = 1, x_2) = \sum_{t=0}^{T_0 - 1} F_{T_n}(t)\mathbb{P}\{T_2 > t|x_2\} > 0$. Since $\Delta_{2,1}$ is strictly increasing in $x_1$, $b_\perp(x_2)$ is the unique point where $\Delta_{2,1}(x_1, x_2)$ crosses 0. Thus, we obtain

$$\rho_{2,1}(x) = \mathbb{1}\{\kappa_2(x_1, x_2) > \kappa_1(x_1, x_2)\} = \mathbb{1}\{\Delta_{2,1}(x_1, x_2) > 0\} = \mathbb{1}\{x_1 > b_\perp(x_2)\}.$$

For any $x_2' > x_2$, $\Delta_{2,1}(b_\perp(x_2), x_2') < \Delta_{2,1}(b_\perp(x_2), x_2) = 0$. $\Delta_{2,1}(b_\perp(x_2'), x_2') = 0$ and $\Delta_{2,1}$ is increasing in $x_1$ implies that $b_\perp(x_2') > b_\perp(x_2)$.

The analysis of $\rho_{3,1}(x)$ follows similar steps. The only difference is that the role of $x_1$ and $x_2$ are reversed since the gap terms in $\Delta_{3,1}(t, x)$ in (25) involve $\mathbb{P}\{T_2 \le t, \mathcal{O}_2|3, x_2\}\mathbb{P}\{T_1 > t|x_1\}$ instead of $\mathbb{P}\{T_1 \le t, \mathcal{O}_1|2, x_1\}\mathbb{P}\{T_2 > t|x_2\}$ and cost is incurred from screening for disease 2. Therefore, we conclude that

$$\rho_{3,1}(x) = \mathbb{1}\{x_2 > b_\perp'(x_1)\}, \tag{33}$$

where $b_\perp'(x_1)$ is increasing in $x_1$.

### B.3. Proof of Lemma 4.3

Recall that

$$\Delta_{4,1}(x_1, x_2) = \sum_{t=0}^{T_0-1} (\mathbb{P}\{T_2 \le t, \mathcal{O}_2|3, x_2\}(\mathbb{P}\{T_1 > t|x_1\} + \mathbb{P}\{T_1 \le t, \mathcal{O}_1|2, x_1\}) + \mathbb{P}\{T_2 > t|x_2\}\mathbb{P}\{T_1 \le t, \mathcal{O}_1|2, x_1\})$$
$$- \lambda c_1(\mathbb{E}[\#_1(\min\{T_0, T_1, t_1^*\})|a, x_1] - 1) - \lambda c_2(\mathbb{E}[\#_2(\min\{T_0, T_2, t_2^*\})|a, x_2] - 1).$$

Fix $x_2 \in [0, 1]$. When $x_1 = 0$,

$$\Delta_{4,1}(x_1 = 0, x_2) = \sum_{t=0}^{T_0-1} \mathbb{P}\{T_2 \le t, \mathcal{O}_2|3, x_2\} - \lambda c_1(\mathbb{E}[\#_1(\min\{T_0, T_1, t_1^*\})|a, x_1 = 0] - 1)$$
$$- \lambda c_2(\mathbb{E}[\#_2(\min\{T_0, T_2, t_2^*\})|a, x_2] - 1).$$

When $x_1 = 1$,

$$\Delta_{4,1}(x_1 = 1, x_2) = \sum_{t=0}^{T_0-1} (\mathbb{P}\{T_2 \le t, \mathcal{O}_2|3, x_2\} + \mathbb{P}\{T_2 > t|x_2\}F_{T_1}(t)) - \lambda c_2(\mathbb{E}[\#_2(\min\{T_0, T_2, t_2^*\})|a, x_2] - 1).$$

Observe that $\Delta_{4,1}(x_1 = 0, x_2) < \Delta_{4,1}(x_1 = 1, x_2)$.

When $x_2 = 0$,

$$\Delta_{4,1}(x_1, x_2 = 0) = \sum_{t=0}^{T_0-1} \mathbb{P}\{T_1 \le t, \mathcal{O}_1|2, x_1\}$$
$$- \lambda c_1(\mathbb{E}[\#_1(\min\{T_0, T_1, t_1^*\})|a, x_1] - 1) - \lambda c_2(S_2 - 1),$$

which is strictly increasing in $x_1$. $\Delta_{4,1}(x_1 = 0, x_2 = 0) = -\lambda c_1(S_1 - 1) - \lambda c_2(S_2 - 1) < 0$, $\Delta_{4,1}(x_1 = 1, x_2 = 0) = \sum_{t=0}^{T_0-1} F_{T_1}(t) - \lambda c_2(S_2 - 1)$. Therefore, when $\sum_{t=0}^{T_0-1} F_{T_1}(t) - \lambda c_2(S_2 - 1) \ge 0$, $b_\equiv(0) \in (0, 1]$. Else, $b_\equiv(0) = +\infty$.

**Proof of** $b_\perp(0) < b_\equiv(0)$**:** Fix $x_2 = 0$. Since $\Delta_{2,1}^T(x_1, x_2 = 0) = \Delta_{4,1}^T(x_1, x_2 = 0)$ and $\Delta_{4,1}^C(x_1, x_2 = 0) > \Delta_{2,1}^C(x_1, x_2 = 0)$, we have $\Delta_{4,1}(x_1, x_2 = 0) < \Delta_{2,1}(x_1, x_2 = 0)$ for all $x_1$. Since $\Delta_{2,1}$ is strictly increasing in $x_1$, this implies that $\Delta_{2,1}$ will cross zero no later than $\Delta_{4,1}$. Hence, $b_\perp(0) < b_\equiv(0)$.

**Proof of** $b_\perp(1) > b_\equiv(1)$**:** Fix $x_2 = 1$. Since $\Delta_{2,1}^T(x_1, x_2 = 1) + \sum_{t=0}^{T_0-1} F_{T_2}(t)(1 - F_{T_1}(t)) \le \Delta_{4,1}^T(x_1, x_2 = 1)$ and $\Delta_{4,1}^C(x_1, x_2 = 1) = \Delta_{2,1}^C(x_1, x_2 = 1)$, we have $\Delta_{4,1}(x_1, x_2 = 1) > \Delta_{2,1}^T(x_1, x_2 = 1)$ for all $x_1$. In Lemma B.3, we showed that $b_\perp(1) < \infty$. Thus, $b_\equiv(1) < \infty$. Since $\Delta_{2,1}$ is strictly increasing in $x_1$, this implies that $\Delta_{2,1}$ will cross zero no earlier than $\Delta_{4,1}$. Hence, $b_\perp(1) > b_\equiv(1)$.

**Proof of** $b_\equiv(x_2)$ **and** $b_\perp(x_2)$ **have a unique intercept in** $x_{2\perp} \in [0, 1]$**:**

Note that $\Delta_{4,1}(x_1 = 0, x_2 = 1) = \sum_{t=0}^{T_0-1} F_{T_2}(t) - \lambda c_1(S_1 - 1)$ and $\Delta_{4,1}(x_1 = 1, x_2 = 1) = \sum_{t=0}^{T_0-1}(F_{T_2}(t) + (1 - F_{T_2}(t))F_{T_1}(t)) > 0$.

(a) Assume that $\Delta_{4,1}(x_1 = 0, x_2 = 1) < 0$. Then, $b_\equiv(1) \in (0, 1)$, and $b_\equiv(1) < b_\perp(1)$. Since $b_\equiv(x_2)$ is decreasing and continuous on $[0, 1]$, one of the following two is possible:

(a.i) There exists $0 \leq x_l < 1$ such that $b_{\equiv}(x_l) = 1$. We also know from Lemma 4.2 that $b_{\perp}(x_l) \in (0, 1)$. Since $b_{\perp}(x_2)$ is increasing, under the assumption that $b_{\equiv}(x_2)$ is decreasing, since both functions $b_{\perp}(x_2)$ and $b_{\equiv}(x_2)$ are continuous (follows from the continuity of $\Delta_{4,1}$ and $\Delta_{4,3}$), $b_{\equiv}(x_2)$ and $b_{\perp}(x_2)$ have a unique intercept in $x_2 \in (x_l, 1)$, say at $x_{2\perp}$.

(a.ii) $b_{\equiv}(0) \leq 1$. Then, uniqueness holds due to the same reasoning as in (i).

(b) Assume that $\Delta_{4,1}(x_1, x_2 = 1) > 0$ for all $x_1 \in [0, 1]$. Then, $b_{\equiv}(1) = -\infty$. Assume that $b_{\equiv}(0) < \infty$. Since $b_{\equiv}(0) \in (0, 1]$, and $b_{\equiv}(x_2)$ is decreasing and continuous, there exists $x_u \in (0, 1)$ such that $b_{\equiv}(x_u) = 0$. We also know that $b_{\perp}(x_u) > 0$. Then, using the same argument as in part $a$, we conclude that $b_{\equiv}(x_2)$ and $b_{\perp}(x_2)$ have a unique intercept in $x_2 \in (0, x_u)$, say at $x_{2\perp}$.

(c) Assume that $b_{\equiv}(1) = -\infty$ and $b_{\equiv}(0) = +\infty$. Then, since $b_{\equiv}(x_2)$ is decreasing and continuous there must be $x_l$ and $x_u$ in $[0, 1]$ such that $b_{\equiv}(x_l) = 1$ and $b_{\equiv}(x_u) = 0$. Since $b_{\perp}(x_l) < 1$ and $b_{\perp}(x_u) > 0$, we conclude that $b_{\equiv}(x_2)$ and $b_{\perp}(x_2)$ have a unique intercept in $x_2 \in (x_l, x_u)$, say at $x_{2\perp}$.

## B.4. Proof of Lemma 4.4

Consider $\rho_{4,3}(x) = \mathbb{1}\{\kappa_4(x) > \kappa_3(x)\}$. Under Assumption B.2, $\Delta_{4,3}(x)$ is strictly increasing in $x_1$.

Fix $x_2 \in [0, 1]$. $\Delta_{4,3}^T(x_1 = 0, x_2) = 0$, $\Delta_{4,3}^C(x_1 = 0, x_2) > 0$. Hence, $\Delta_{4,3}(x_1 = 0, x_2) < 0$. $\Delta_{4,3}^T(x_1 = 1, x_2) > 0$, $\Delta_{4,3}^C(x_1 = 1, x_2) = 0$. Hence, $\Delta_{4,3}(x_1 = 1, x_2) > 0$. Since $\Delta_{4,3}(x_1, x_2)$ is strictly increasing in $x_1$, there is a unique zero crossing, which is denoted by $x_1 = b_{\top}(x_2)$.

**Proof of $b_{\perp}(0) = b_{\top}(0) = b_{\top}(1)$:** Note that

$$\Delta_{4,3}^T(x_1, x_2 = 0) = \Delta_{4,3}^T(x_1, x_2 = 1) = \Delta_{2,1}^T(x_1, x_2 = 0) = \sum_{t=0}^{T_0-1} \mathbb{P}\{T_1 \leq t, \mathcal{O}_1|2, x_1\}, \tag{34}$$

$$\Delta_{4,3}^C(x_1, x_2 = 0) = \Delta_{4,3}^C(x_1, x_2 = 1) = \Delta_{2,1}^C(x_1, x_2 = 0),$$

which implies that $\Delta_{4,3}(x_1, x_2 = 0) = \Delta_{4,3}(x_1, x_2 = 1) = \Delta_{2,1}(x_1, x_2 = 0)$. Therefore, we conclude that they have the same zero crossing, i.e., $b_{\perp}(0) = b_{\top}(0) = b_{\top}(1)$.

**Proof of $b_{\perp}(0) \leq b_{\top}(x_2) \leq b_{\perp}(x_2)$:** $\Delta_{4,3}^T(x_1, x_2) \geq \Delta_{2,1}^T(x_1, x_2)$ and $\Delta_{4,3}^C(x_1, x_2) = \Delta_{2,1}^C(x_1, x_2)$, which implies that $\Delta_{4,3}(x_1, x_2) \geq \Delta_{2,1}(x_1, x_2)$. Hence, $b_{\top}(x_2) \leq b_{\perp}(x_2)$. From (34), note that $\Delta_{2,1}^T(x_1, x_2 = 0) \geq \Delta_{4,3}^T(x_1, x_2)$ and $\Delta_{2,1}^C(x_1, x_2 = 0) = \Delta_{4,3}^C(x_1, x_2)$, which implies that $\Delta_{2,1}(x_1, x_2 = 0) \geq \Delta_{4,3}(x_1, x_2)$. Hence, $b_{\top}(x_2) \geq b_{\perp}(0)$.

**Proof of $b_{\top}(0) = b_{\top}(1) \leq b_{\top}(x_2)$:** This again follows from the fact that $\Delta_{4,3}(x_1, x_2 = 0) = \Delta_{4,3}(x_1, x_2 = 1) \geq \Delta_{4,3}(x_1, x_2)$, so the zero crossing of the latter cannot be earlier than the others.

## B.5. Proof of Proposition 4.1

To construct $\rho_1^*(x)$, we consider all cases separately, noting that the values of $\kappa$ corresponding to actions where the first disease is screened ($a = 2$ or $a = 4$) should be greater than those corresponding to actions where the first disease is not screened ($a = 1$ or $a = 3$):

| $\kappa_2(x) > \kappa_1(x)$ | $\kappa_4(x) > \kappa_1(x)$ | $\kappa_4(x) > \kappa_3(x)$ | $\rho_1^*(x)$ |
|---|---|---|---|
| 0 | 0 | 0 | 0 |
| 0 | 0 | 1 | 0 |
| 0 | 1 | 0 | 0 |
| 0 | 1 | 1 | 1 |
| 1 | 0 | 0 | $\times$ |
| 1 | 0 | 1 | 1 |
| 1 | 1 | 0 | $\times$ |
| 1 | 1 | 1 | 1 |

For example, in line 2, we have $\kappa_2(x) < \kappa_1(x)$ and $\kappa_4(x) < \kappa_1(x)$, which means that neither of the actions that screen disease 1 is activated since neither $\kappa_2(x)$ or $\kappa_4(x)$ is the largest, leading to $\rho_1^*(x) = 0$. In line 3, we have $\kappa_2(x) < \kappa_1(x)$ and $\kappa_1(x) < \kappa_4(x)$, but $\kappa_4(x) < \kappa_3(x)$, again meaning that $\rho_1^*(x) = 0$. But, for instance, in line 6, since we have $\kappa_2(x) > \kappa_1(x), \kappa_1(x) > \kappa_4(x), \kappa_4(x) > \kappa_3(x)$, this time $\rho_1^*(x) = 1$ since $\kappa_2(x)$ is the largest.

It should be observed that lines 5 and 7 are inconsistent since $\Delta_{4,3}^T(x) \geq \Delta_{2,1}^T(x)$ and $\Delta_{4,3}^C(x) = \Delta_{2,1}^C(x)$, hence $\Delta_{4,3}(x) \geq \Delta_{2,1}(x)$. Therefore, $\kappa_2(x) > \kappa_1(x)$ implies $\kappa_4(x) > \kappa_3(x)$, hence we cannot have $\kappa_2(x) > \kappa_1(x)$ and $\kappa_4(x) < \kappa_3(x)$ at the same time.

As a result, we can write the relation as:

$$\rho_1^*(x) \equiv \rho_{2,1}(x) \vee (\rho_{4,1}(x) \wedge \rho_{4,3}(x)). \tag{35}$$

Then it is enough for the optimal referral decision boundary to satisfy $\kappa_2(x) > \kappa_1(x)$, which will imply that $\kappa_4(x) > \kappa_3(x)$. This guarantees that either $\kappa_2(x)$ or $\kappa_4(x)$ is larger than both $\kappa_1(x)$ and $\kappa_3(x)$, because it is impossible for $\kappa_1(x)$ and $\kappa_3(x)$ to be the largest, hence the activation for the first disease is guaranteed. For example, if $\kappa_3(x) > \kappa_2(x)$, then $\kappa_4(x)$ will be the largest. If $\kappa_2(x) > \kappa_1(x)$ is not true, then we should have both $\kappa_4(x) > \kappa_3(x)$ and $\kappa_4(x) > \kappa_1(x)$, which will make $\kappa_4(x)$ the largest.

Using symmetry for the second disease, we have,

$$\rho_2^*(x) \equiv \rho_{3,1}(x) \vee (\rho_{4,1}(x) \wedge \rho_{4,2}(x)), \tag{36}$$

where

$$\rho_{3,1}(x) = \mathbb{1}\{\ \kappa_3(x) > \kappa_1(x)\ \},$$
$$\rho_{4,1}(x) = \mathbb{1}\{\ \kappa_4(x) > \kappa_1(x)\ \},$$
$$\rho_{4,2}(x) = \mathbb{1}\{\ \kappa_4(x) > \kappa_2(x)\ \}.$$

### B.6. Proof of Proposition 4.5

Fix $x_2 \in (0, 1)$. Note that

$$\Delta_{4,1}^T(x_1, x_2) - \Delta_{4,3}^T(x_1, x_2) = \sum_{t=0}^{T_0-1} \mathbb{P}\{T_1 > t | x_1\} \mathbb{P}\{T_2 \leq t, \mathcal{O}_2 | 3, x_2\}, \tag{37}$$

is strictly decreasing in $x_1$, and $\Delta_{4,1}^C(x_1, x_2) - \Delta_{4,3}^C(x_1, x_2) = c_2(\mathbb{E}[\#_2(\min\{T_0, T_2, t_2^*\}) | a, x] - 1)$ is constant in $x_1$. Therefore, $\Delta_{4,1}(x_1, x_2) - \Delta_{4,3}(x_1, x_2)$ is strictly decreasing in $x_1$.

Fix $x_2 = 0$. Then,

$$\Delta_{4,1}^T(x_1, x_2 = 0) = \sum_{t=0}^{T_0-1} \mathbb{P}\{T_1 \leq t, \mathcal{O}_1 | 2, x_1\},$$

is strictly increasing in $x_1$, $\Delta_{4,1}^C(x_1, x_2 = 0)$ is non-increasing in $x_1$. Therefore, $\Delta_{4,1}(x_1, x_2 = 0)$ is strictly increasing in $x_1$. Also note that $\Delta_{4,1}^T(x_1, x_2 = 0) = \Delta_{4,3}^T(x_1, x_2 = 0)$ and $\Delta_{4,1}^C(x_1, x_2 = 0) - \Delta_{4,3}^C(x_1, x_2 = 0) = c_2(S_2 - 1) > 0$. Therefore, $\Delta_{4,1}(x_1, x_2 = 0) < \Delta_{4,3}(x_1, x_2 = 0)$. Due to this, and strictly increasing nature of $\Delta_{4,1}(x_1, x_2 = 0)$, its zero crossing (if any) should happen after the zero crossing of $\Delta_{4,3}(x_1, x_2 = 0)$. Hence, it holds that $b_\equiv(0) > b_\top(0)$.

Fix $x_2 = 1$. $\Delta_{4,1}^C(x_1, x_2 = 1) = \Delta_{4,3}^C(x_1, x_2 = 1) = c_1(\mathbb{E}[\#_1(\min T_0, T_1, t_1^*) | a, x] - 1)$, $\Delta_{4,1}^T(x_1, x_2 = 1) - \Delta_{4,3}^T(x_1, x_2 = 1) > 0$ by (37). Therefore, $\Delta_{4,1}^T(x_1, x_2 = 1) > \Delta_{4,3}^T(x_1, x_2 = 1)$. Since $\Delta_{4,3}(x_1, x_2 = 1)$ is strictly increasing in $x_1$, its zero crossing should happen later than the zero crossing (if any) of $\Delta_{4,1}(x_1, x_2 = 0)$. Hence, it holds that $b_\equiv(1) < b_\top(1)$.

Under the assumption that $b_\equiv(x_2)$ and $b_\top(x_2)$ have a unique intercept, the continuity of these functions (which results from the continuity of $\Delta$ functions) imply that there exists $x_{2\top} \in [0, 1]$, where $b_\top(x_2) \geq b_\equiv(x_2)$ for $x_2 \geq x_{2\top}$. Moreover, since $b_\top(x_2) \leq b_\perp(x_2)$ for all $x_2$, this value should be greater than or equal to $x_{2\perp}$, where $b_\perp(x_2)$ exceeds $b_\equiv(x_2)$.

Based on this, we first combine these two boundaries due to the parentheses between $\rho_{4,1}(x)$ and $\rho_{4,3}(x)$ in Lemma 4.1. Due to the "and" condition, points must lie within both regions (to satisfy both of them). Thus, we take the larger boundary. Let us introduce an intermediate boundary as:

$$b_\wedge(x_2) = \begin{cases} b_\equiv(x_2) & \text{if} \quad x_2 \in (0, x_{2\top}) \\ b_\top(x_2) & \text{if} \quad x_2 \in [x_{2\top}, 1] \end{cases} \tag{38}$$

Now, we can combine this with the boundary of $\rho_{2,1}(x)$ to obtain the boundary $\zeta_1(x_2)$ for the optimal rule $\rho_1^*(x)$. Given the "or" condition between in $\rho_{2,1}(x)$ and $\rho_{4,1}(x) \wedge \rho_{4,3}(x)$ (whose boundary we just defined as $b_\wedge(x_2)$) in Lemma 4.1, we select the smaller boundary value of the two, as it suffices for any point to lie within at least one of the two regions. Starting from $x_2 = 0$, we initially use the lower boundary $b_\perp(x_2)$, since we have shown that $b_\perp(x_2) < b_\equiv(x_2)$ for $x_2 < x_{2\perp}$ in Lemma 4.3. When $b_\perp(x_2)$ exceeds $b_\equiv(x_2)$ at $x_{2\perp}$, the boundary switches to $b_\equiv(x_2)$, which is now smaller. We have also shown that $b_\top(x_2) \leq b_\perp(x_2)$ for all $x_2 \in [0, 1]$ in Lemma 4.4, thus we conclude that $b_\perp(x_2) \geq b_\wedge(x_2)$ for $x_2 \geq x_{2\perp}$, which leads to our optimal referral decision boundary $\zeta_1(x_2)$ being obtained as:

$$\zeta_1(x_2) = \begin{cases} b_\perp(x_2) & \text{if} \quad x_2 \in [0, x_{2\perp}] \\ b_\wedge(x_2) & \text{if} \quad x_2 \in (x_{2\perp}, 1] \end{cases} \tag{39}$$

given that $x_{2\perp} \leq x_{2\top}$ as we have just shown. Finally, by substituting $b_\wedge(x_2)$, our optimal referral decision $\rho_1^*(x) = \mathbb{1}\{x_1 > \zeta_1(x_2)\}$ is obtained as:

$$\zeta_1(x_2) = \begin{cases} b_\perp(x_2) & \text{if} \quad x_2 \in [0, x_{2\perp}] \\ b_\equiv(x_2) & \text{if} \quad x_2 \in (x_{2\perp}, x_{2\top}) \\ b_\top(x_2) & \text{if} \quad x_2 \in [x_{2\top}, 1] \end{cases} \tag{40}$$

Moreover, $\zeta_1(0) = \zeta_1(1)$ since we have shown that $b_\perp(0) = b_\top(1)$ (by Lemma 4.4). Also observe that $\zeta_1(0) = b_\perp(0) \leq b_\top(x_2) \leq b_\perp(x_2)$ for all $x_2 \in [0, 1]$ (by Lemma 4.4), and also $\zeta_1(0) = b_\perp(0) = b_\top(1) \leq b_\top(x_{2\top}) = b_\equiv(x_{2\top}) \leq b_\equiv(x_2)$ for $x_2 \leq x_{2\top} \leq 1$ (by Lemma 4.3 and 4.4, using that $b_\equiv(x_2)$ is non-increasing). Consequently, $\zeta_1(0) \leq b_\equiv(x_2)$ for $x_2 \in (x_{2\perp}, x_{2\top})$. Combining all of these, we finally conclude $\zeta_1(0) \leq \zeta_1(x_2)$ for all $x_2 \in [0, 1]$.

Due to the symmetry of the functions by replacing the index $n = 1$ with $n = 2$ and vice-versa, and consequently due to the symmetry of the boundaries, for the second disease, we also obtain the form $\rho_2^*(x) = \mathbb{1}\{x_2 > \zeta_2(x_1)\}$ where

$$\zeta_2(x_1) = \begin{cases} b'_\perp(x_1) & \text{if} \quad x_1 \in [0, x_{1\perp}] \\ b_\equiv(x_1) & \text{if} \quad x_1 \in (x_{1\perp}, x_{1\top}) \\ b'_\top(x_1) & \text{if} \quad x_1 \in [x_{1\top}, 1] \end{cases} \tag{41}$$

### B.7. Proof of Proposition 4.6

Recall that, in our case for two diseases, the independent referral decisions will have the form $\bar\rho(X) = [\bar\rho_1(x_1), \bar\rho_2(x_2)]^\top$, where $\bar\rho_n$'s are independent from $x_i, \forall i \neq n$. In this case, redefine action space for each disease as $b : \bar\rho_n(x_n) \longrightarrow \{1, 2\}$, where $b = 1$ corresponds to the case where disease $n$ is not being screened and $b = 2$ corresponds to the case where disease $n$ is screened. Unlike action $a$ in joint screening, action $b$ only depends on the risk $x_n$ of disease $n$.

Recall that $\mathcal{U}_n = \{\Theta_n = 1\} \wedge \{t_n^* \geq T_n\}$. In independent screening, the decision to screen for disease $n$ depends only on the variables related to disease $n$. Let $T_n^* := T_n \mathbb{1}\{\mathcal{U}_n\} + T_0 \mathbb{1}\{\neg\mathcal{U}_n\}$. The independent screening problem for disease $n$ is formulated as

$$\text{maximize}_{\pi \in \Pi_n} \quad \mathbb{E}[\min\{T_0, T_n^*\}] \quad \text{subject to} \quad \mathbb{E}[\#_n(\min\{T_0, T_n, t_n^*\})] \leq B_n, \tag{42}$$
$$\mathbb{P}\{\Theta_n = 0, \hat\theta_n(t) = 1\} \leq \alpha_n, \forall t \in [T_0],$$

where $B_n$ is the budget reserved for screening of disease $n$, and $\Pi_n$ represents the set of all screening and diagnosis policies that is based only on the risk and observations from disease $n$.

Let action $b = 1$ represent no screening and $b = 2$ represent screening for disease $n$. Using posterior probability of having the disease with the calibrated threshold $\gamma_n$ as the diagnosis rule, under the uniform risk profile, the optimization problem for the independent referral decision of disease $n$ becomes

$$\text{maximize}_{q_n} \int_{x \in \mathcal{X}} \left( \sum_{b=1}^{2} q_n(b|x) r_{n,b,x} \right) dx \tag{43}$$

$$\text{subject to} \quad \begin{cases} \displaystyle\int_{x \in \mathcal{X}} \left( \sum_{b=1}^{2} q_n(b|x) m_{n,b,x} \right) dx \leq B_n, \\ \\ q_n(b|x) \geq 0, \forall b \in [2], x \in \mathcal{X}, \displaystyle\sum_{b=1}^{2} q_n(b|x) = 1, \forall x \in \mathcal{X}, \end{cases}$$

where $r_{n,b,x} = \mathbb{E}[\min\{T_0, T_n^*\}|b, x]$ and $m_{n,b,x} := c_n \mathbb{E}[\#_n|b, x]$. The expected costs satisfy the following: $m_{n,1,x} = 0$, $m_{1,2,x} = m_{2,x}, m_{2,2,x} = m_{3,x}, m_{1,2,x} + m_{2,2,x} = m_{4,x} = m_{2,x} + m_{3,x}$.

To characterize the structure of $\bar{\rho}_n^*(x_n)$, we proceed with the Lagrange dual problem associated with (43). In particular,

$$\bar{\rho}_n^*(x_n) = \mathbb{1}\{ \mathbb{E}[\min\{T_0, T_n^*\}|b = 2, x_n] - \lambda c_n \mathbb{E}[\#_1|b = 2, x_n] > \mathbb{E}[\min\{T_0, T_n^*\}|b = 1, x_n] - \lambda c_n \mathbb{E}[\#_1|b = 1, x_n] \},$$

where $\lambda > 0$ depends on $B_n$. Basically, disease $n$ will be screened when $\kappa_2$ (corresponding to screening) exceeds $\kappa_1$ (corresponding to not screening).

The survival functions for independent screening of disease $n$ are:

$$S_{T_n^*}(t, x_n, b = 2) = \mathbb{P}\{T_n > t|x_n\} + \mathbb{P}\{T_n \leq t, \mathcal{O}_n|b = 2, x_n\}, \quad S_{T_n^*}(t, x_n, b = 1) = \mathbb{P}\{T_n > t|x_n\}.$$

Then, $S_{T_n^*}(t, x_n, b = 2) - S_{T_n^*}(t, x_n, b = 1) = \mathbb{P}\{T_n \leq t, \mathcal{O}_n|b = 2, x_n\} \geq 0$. Note that $\mathbb{E}[\min\{T_0, T_n^*\}|b = 2, x] - \mathbb{E}[\min\{T_0, T_n^*\}|b = 1, x] = \sum_{t=0}^{T_0-1} \mathbb{P}\{T_n \leq t, \mathcal{O}_n|b = 2, x_n\}$ is increasing in $x_n$. Moreover, $\mathbb{E}[\#_n|b = 2, x_n]$ is non-increasing in $x_n$ and $\mathbb{E}[\#_n|b = 1, x_n] = 0$. Therefore, $\kappa_2(x_n) - \kappa_1(x_n)$ is increasing in $x_n$.

For $x_n = 0$, $\mathbb{E}[\min\{T_0, T_n^*\}|b = 2, x_n] = \mathbb{E}[\min\{T_0, T_n^*\}|b = 1, x_n]$ and $\mathbb{E}[\#_1|b = 2, x_n] > 0$. Therefore, $\kappa_2(x_n = 0) < \kappa_1(x_n = 0)$. For $x_n = 1$, $\mathbb{E}[\min\{T_0, T_n^*\}|b = 2, x_n] > \mathbb{E}[\min\{T_0, T_n^*\}|b = 1, x_n]$ and $\mathbb{E}[\#_1|b = 2, x_n] = 0$. Therefore, $\kappa_2(x_n = 1) > \kappa_1(x_n = 1)$.

Since $\kappa_2(x_n) - \kappa_1(x_n)$ is increasing in $x_n$, there exists a unique point $\bar{\zeta}_n \in (0, 1)$ such that $\kappa_2(x_n) - \kappa_1(x_n) \leq 0$ for $x_n \leq \bar{\zeta}_n$ and $\kappa_2(x_n) - \kappa_1(x_n) \geq 0$ for $x_n \geq \bar{\zeta}_n$. Assuming that the functions $\kappa_2(x_n)$ and $\kappa_1(x_n)$ are continuous in $(0, 1)$, $\kappa_2(\bar{\zeta}_n) - \kappa_1(\bar{\zeta}_n) = 0$.

Hence we obtain the form $\bar{\rho}_1^*(x_1) = \mathbb{1}\{x_1 > \bar{\zeta}_1\}$ and $\bar{\rho}_2^*(x_2) = \mathbb{1}\{x_2 > \bar{\zeta}_2\}$, where $\bar{\zeta}_1 \in (0, 1)$ and $\bar{\zeta}_2 \in (0, 1)$ are constants that depend on the screening budgets reserved for each disease.

We analyze the behavior of the policies/referral decisions under constant thresholds. First, observe the following fundamental relationship under these independent referral decisions:

- Higher thresholds result in fewer screenings, reducing the overall cost.

- Lower thresholds result in more screenings, improving survival times.

Given these observations, the optimal constant-threshold referral decision corresponds to the threshold $\bar{\zeta}_1$ that exactly satisfies the budget constraint. Thresholds below this value violate the budget as they involve higher screening costs, while thresholds above it underutilize the budget, leading to suboptimal survival times.

Consider the case when the unified budget $B$ is split among two disease screening programs as in the optimal unified referral decision. Now, consider the extreme cases of constant thresholds:

- At $\zeta_1(0) = \min_{x_2 \in [0,1]} \zeta_1(x_2)$, the threshold is low, implying that this referral decision screens strictly more people compared to the overall optimal referral decision of $\bar{\zeta}_1$. As a result, $\zeta_1(0)$ violates the budget due to excessive cost.

- At $\max_{x_2 \in [0,1]} \zeta_1(x_2)$, the threshold is high, resulting in strictly fewer screenings. Since this referral decision has a lower cost than $\bar{\zeta}_1$, it is under budget.

The key insight is that $\bar{\zeta}_1$, the overall optimal referral decision, balances survival time and cost such that it satisfies the budget exactly. By continuity of the cost function with respect to the threshold, and since one extreme ($\zeta_1(0)$) exceeds the budget while the other ($\max_{x_2 \in [0,1]} \zeta_1(x_2)$) remains under budget, it follows from the Intermediate Value Theorem that $\bar{\zeta}_1$ must lie between these two extremes. This establishes the result:

$$\zeta_1(0) = \min_{x_2 \in [0,1]} \zeta_1(x_2) \leq \bar{\zeta}_1 \leq \max_{x_2 \in [0,1]} \zeta_1(x_2).$$

## C. Empirical Justification of Theoretical Assumptions

The assumptions that $b_{\equiv}(x_2)$ is decreasing in $x_2$, and that $b_{\equiv}(x_2)$ and $b_{\top}(x_2)$ have a unique intercept, are difficult to prove analytically. This complexity arises from the dependence of the decision boundaries on the prior risk profiles in addition to the likelihood. To provide empirical support for these assumptions, we conduct comprehensive simulations under varying diagnosis thresholds $\gamma$, different values $\lambda$ to model the tradeoff between survival and cost, and alternative sampling distributions for the observations $y$, including both Bernoulli and Gaussian noise models. Screening costs and screening periods are set to 1 for both diseases.

In Figures 6, 7, and 8, we plot $b_{\equiv}$ and $b_{\top}$ for different values of $\lambda$ under Gaussian and Bernoulli likelihoods. The adverse event times $T_1$ and $T_2$ are assumed to be equal and deterministic. Under Gaussian likelihood, the mean is zero when no disease is present and 1 when the disease is present. The noise variance is 1. Under Bernoulli likelihood, the probability of observing 1 when a disease is present is $p$, while when not present, it is $q$. These plots demonstrate that under reasonable conditions, the assumptions hold approximately in practice, and the observed boundaries intersect uniquely.

To investigate the effects of the randomness of adverse event times on the decision boundaries, we also performed simulations under Poisson distributed and uniformly distributed adverse event times (Figures 9 and 10).

These results suggest that although analytical guarantees may be elusive, the structure of the optimal boundaries observed in practice largely adheres to the assumptions stated in Proposition 4.5. When violations do occur, they appear minimal and do not disrupt the overall interpretability of the policies.

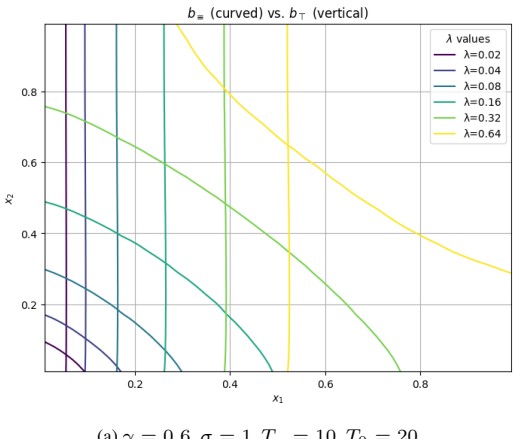
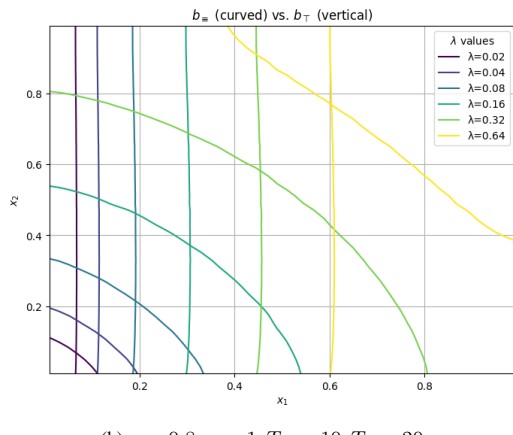

(a) $\gamma = 0.6, \sigma = 1, T_n = 10, T_0 = 20$        (b) $\gamma = 0.8, \sigma = 1, T_n = 10, T_0 = 20$

Figure 6: Decision boundaries for Gaussian activations (Set 1)

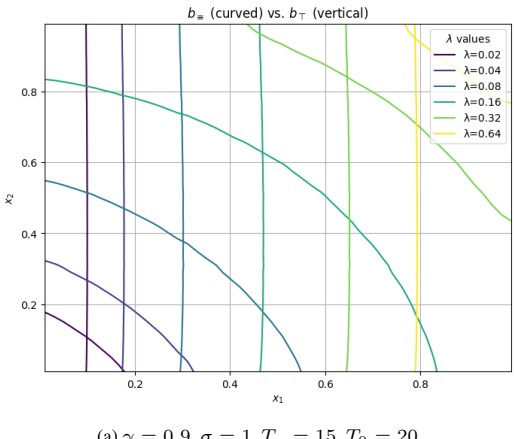
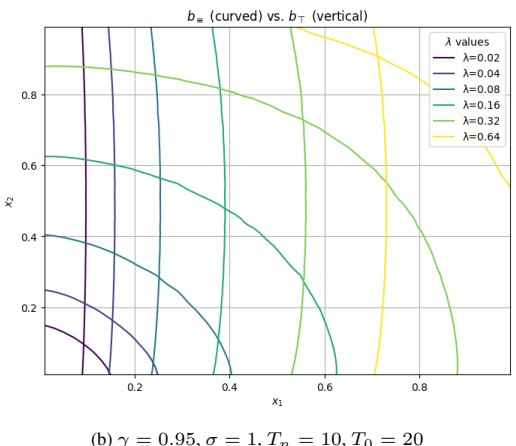

(a) $\gamma = 0.9, \sigma = 1, T_n = 15, T_0 = 20$        (b) $\gamma = 0.95, \sigma = 1, T_n = 10, T_0 = 20$

Figure 7: Decision boundaries under Gaussian likelihood.

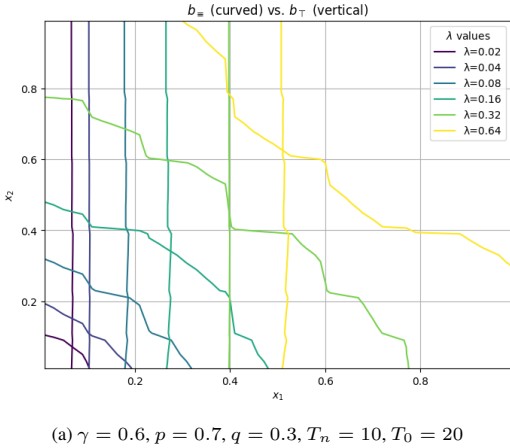

(a) $\gamma = 0.6$, $p = 0.7$, $q = 0.3$, $T_n = 10$, $T_0 = 20$

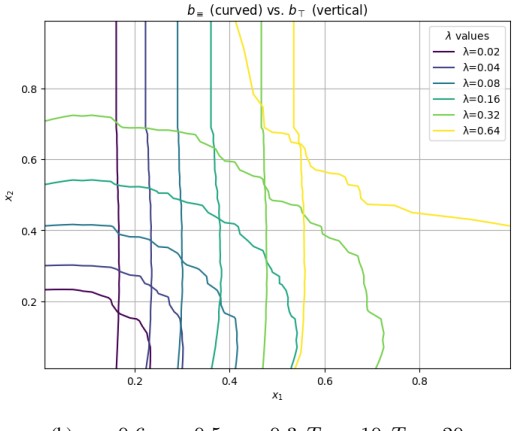

(b) $\gamma = 0.6$, $p = 0.5$, $q = 0.3$, $T_n = 10$, $T_0 = 20$

Figure 8: Decision boundaries under Bernoulli likelihood.

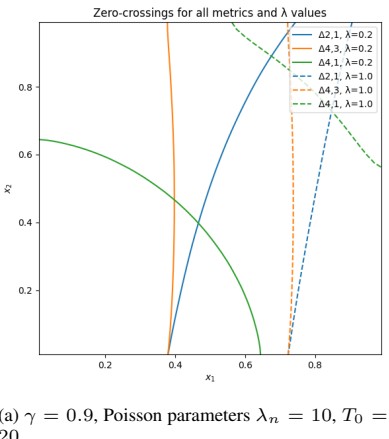

(a) $\gamma = 0.9$, Poisson parameters $\lambda_n = 10$, $T_0 = 20$

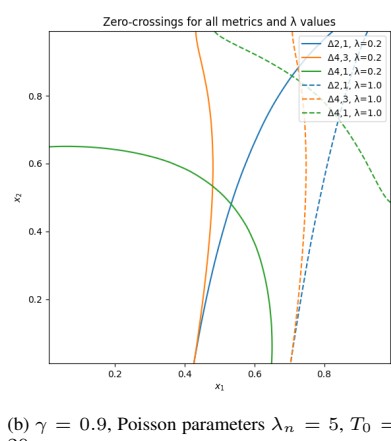

(b) $\gamma = 0.9$, Poisson parameters $\lambda_n = 5$, $T_0 = 20$

Figure 9: Decision boundaries for Poisson distributed adverse event times and Gaussian likelihood.

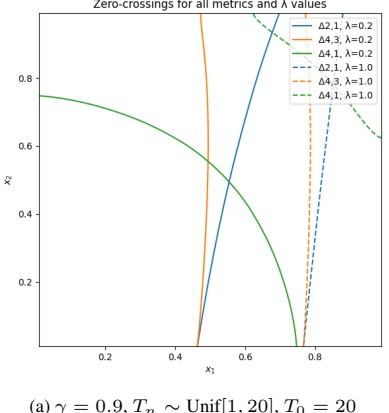

(a) $\gamma = 0.9$, $T_n \sim \text{Unif}[1, 20]$, $T_0 = 20$

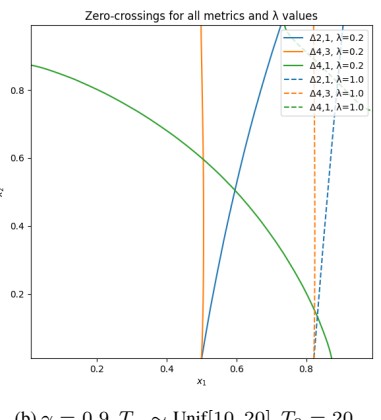

(b) $\gamma = 0.9$, $T_n \sim \text{Unif}[10, 20]$, $T_0 = 20$

Figure 10: Decision boundaries for uniformly distributed adverse event times and Gaussian likelihood.

# D. Supplementary Experiments

Figure 11 presents a plot of the actions/referral decisions based on the solution provided by CVXPY, alongside the boundaries derived in Section 4, for better visualization.

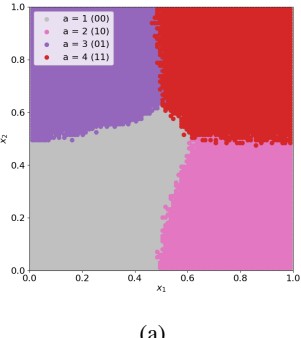
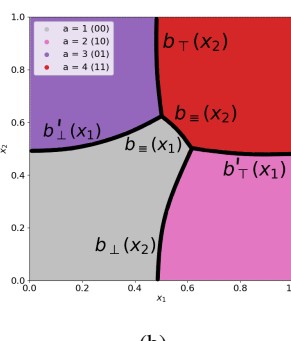

(a)                                         (b)

Figure 11: (a) Optimal boundaries for the actions, as provided by CVXPY for $B = 10$, (b) Matching with the boundaries derived in Section 4.

## D.1. Varying Key Parameters $(B, c_n, \mu_n, \alpha, \beta)$

A series of simulations was conducted to evaluate the policy framework's performance and accuracy, with systematic variations in key parameters. Findings across different configurations include the following:

**Increasing the Budget $B$:** An increase in the budget $B$ allowed for more frequent screening, effectively reducing the "grey region" (where no specific action was taken due to cost limitations, see Figure 12). As a result, the regions implementing active screening policies ($a = 2, 3, 4$) expanded, illustrating that higher budgets enable more comprehensive screening and reduce the risk of undiagnosed conditions.

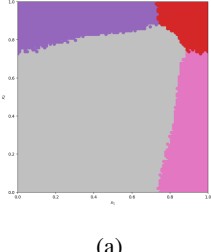
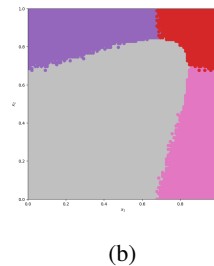
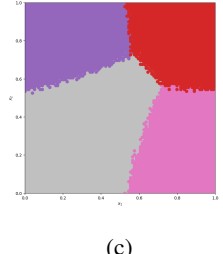
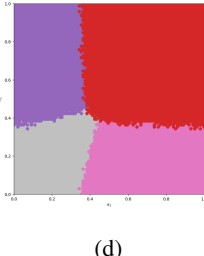

(a)                    (b)                    (c)                    (d)

Figure 12: Varying the budget ($B$) with other parameters fixed, (a) $B = 2$, (b) $B = 4$, (c) $B = 7$, (d) $B = 15$.

**Varying Mean Adverse Event Times ($\mu_n = \mathbb{E}[T_n]$) for Diseases:** When the mean time to adverse events was shorter for a specific disease, the model showed a tendency to prioritize screening for that disease, aligning with the objective of early detection. Diseases with longer expected event times were deprioritized, as immediate screening was less urgent (see Figure 13).

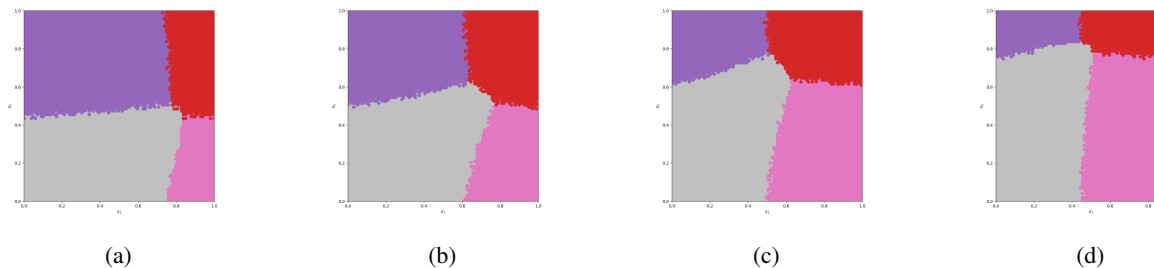

Figure 13: Varying the mean adverse event times $(T_n)$ for the diseases (where they are distributed with Gaussian distribution) with other parameters fixed, (a) $\mu_1 = 37, \mu_2 = 30.1$, (b) $\mu_1 = 34, \mu_2 = 30.1$, (c) $\mu_1 = 30.1, \mu_2 = 34$, (d) $\mu_1 = 30.1, \mu_2 = 37$.

**Modifying Feature Distribution Parameters ($\alpha$ and $\beta$):** As $\alpha$ is decreased and $\beta$ is increased, the probabilities $p(x)$ associated with higher-risk regions requiring screening decrease. This reduction enables more extensive screening, as a greater number of points satisfy the cost constraint. Consequently, the grey regions in the plot, which represent unscreened areas, shrink in size (see Figure 14).

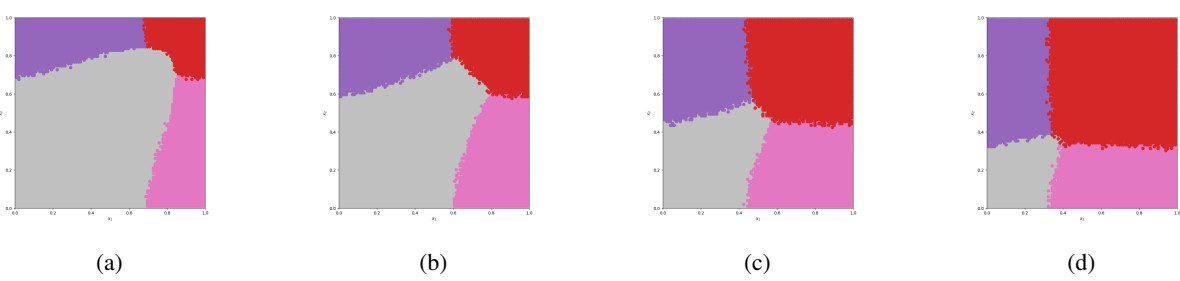

Figure 14: Varying the coefficients $\alpha$ and $\beta$ in the PDF of the feature distribution $x$ with other parameters fixed, (a) $\alpha = 5, \beta = 1$, (b) $\alpha = 5, \beta = 3$, (c) $\alpha = 3, \beta = 3$, (d) $\alpha = 1, \beta = 5$.

### D.2. Decision Boundaries Based on $\kappa_i$'s and Comparison

After conducting simulations with the proposed optimizations, we plotted $\kappa_i$'s for each action to assess its behavior across the feature dimensions $x_1$ and $x_2$. To approximate the expected value, we employed Monte Carlo integration. We then compared the values of $\kappa_i$ across the four actions, and selected the action associated with the highest $\kappa_i$ at each point $(x_1, x_2)$. This allowed us to delineate decision boundaries by identifying the action that maximizes $\kappa_i$ as a function of patient features $x$. In the mathematical proof, we also derived the optimal referral decisions $\rho_n(x)$ by determining the boundaries at which one parameter $\kappa_i$ surpasses the others. This approach is based on the principle that the optimal policy corresponds to the parameter $\kappa_i$ that achieves the maximum value.

Plots of $\kappa_i$ for each action (see Figure 15), with $\lambda = 0.25$, demonstrate patterns consistent with our analytical results. Specifically: (a) without screening, $\kappa_1$ decreases with both $x_1$ and $x_2$, peaking at $(x_1 = 0, x_2 = 0)$; (b) when screening the first disease, $\kappa_2$ decreases with $x_2$ but increases with $x_1$, peaking at $(x_1 = 1, x_2 = 0)$; (c) when screening only the second disease, $\kappa_3$ decreases with $x_1$ but increases with $x_2$, peaking at $(x_1 = 0, x_2 = 1)$; and (d) when both diseases are screened, $\kappa_4$ increases with both $x_1$ and $x_2$, peaking at $(x_1 = 1, x_2 = 1)$. These observations show peak values at boundary points where $x_i$ is either 1 or 0, indicating that high-risk values minimize cost, while zero-risk values maximize survival time.

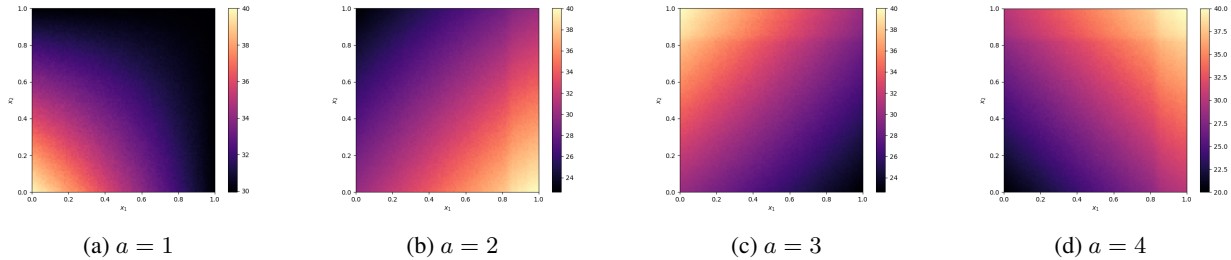

(a) $a = 1$      (b) $a = 2$      (c) $a = 3$      (d) $a = 4$

Figure 15: Plots of $\kappa_i$ for each action with $\lambda = 0.25$, observe the behaviors which match our mathematical results. (a) When no screening is made, $\kappa_1$ is decreasing in both $x_1$ and $x_2$, with the peak value occurring at $(x_1 = 0, x_2 = 0)$. (b) When the first disease is screened, $\kappa_2$ is decreasing in $x_2$ but increasing in $x_1$, with the peak value occurring at $(x_1 = 1, x_2 = 0)$. (c) When the second disease is screened, $\kappa_3$ is decreasing in $x_1$ but increasing in $x_2$, with the peak value occurring at $(x_1 = 0, x_2 = 1)$. (d) When both diseases are screened, $\kappa_4$ is increasing in both $x_1$ and $x_2$, with the peak value occurring at $(x_1 = 1, x_2 = 1)$.

Next, we examined the values of $\kappa_i$ across the four actions, selecting the action with the highest $\kappa_i$ at each point. We generated decision boundaries by selecting actions with the largest $\kappa$ values, and recreated the decision points found in our mathematical proof by locating intersections where one $\kappa_i$ surpasses the others (see Figure 16b). A comparison of these boundaries, derived by selecting the maximum $\kappa_i$ (per our theoretical framework) and those from CVXPY's linear programming solution, confirms that the boundary shapes are nearly identical. This validates our proof's credibility and alignment with the convex optimization results (see Figure 17).

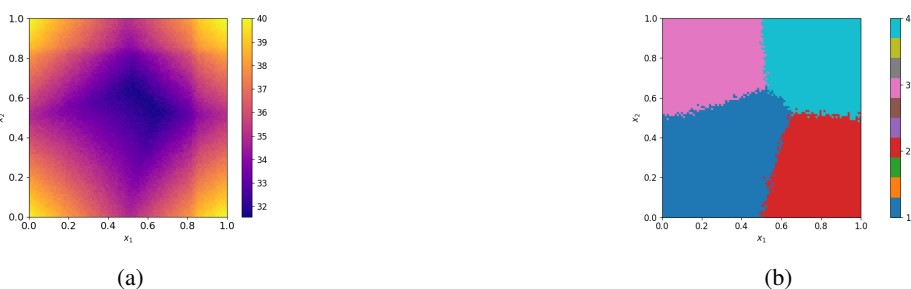

(a)                (b)

Figure 16: (a) Maximum $\kappa_i$ values among the four actions $a = 1, 2, 3, 4$. The maximum values of $\kappa_i$ is plotted by comparing the four values for each $(x_1, x_2)$. The peak values occur at the corners where $x_i$'s are either 1 or 0, since having the risk as 1 minimizes the cost while having no risk maximizes the survival time. (b) The decision boundary obtained by choosing the action with the largest $\kappa$ value. This is essentially what we did in the proof, we formed the boundaries by finding the intersection points at which one of the $\kappa_i$'s exceeds the other.

**D.3. Supplementary Material for Comparison with Independent Screening**

We present the plots of chosen actions for independent screening and unified screening in Figure 18 for better visualization. We determine the survival times ($\mathbb{E}\left[\min\{T_0, T^*\}|a, x\right]$) for the first and second diseases separately, and by choosing the minimum of these two survival times (as the adverse event that occurs first is considered, like in our joint model), we obtain the plot of survival times for the independent model (see Figure 19a). We also created a survival time plot for our joint screening model (see Figure 19b).

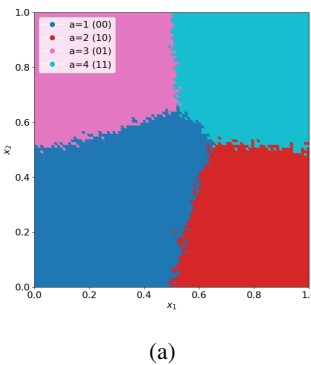
(a)

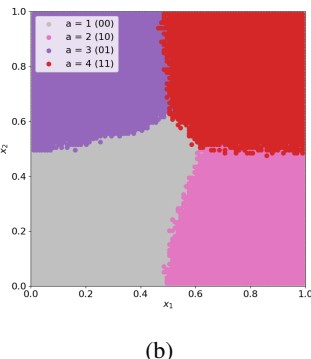
(b)

Figure 17: Comparison of boundaries (a) obtained by choosing the action/referral with the largest value of $\kappa_i$ (as we did in the mathematical proof) and (b) by solving as linear programming by CVXPY. Observe that the boundary shapes are almost identical, verifying the credibility of our proof and how it aligns with the convex optimization solution provided by CVXPY.

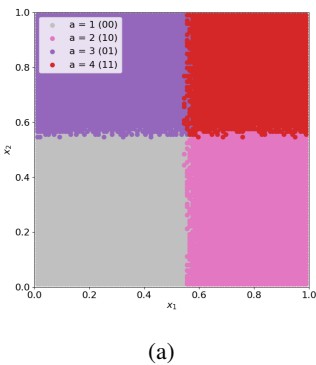
(a)

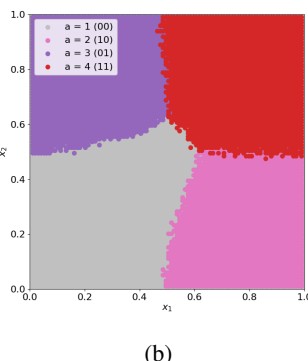
(b)

Figure 18: Comparison of chosen actions (a) for independent screening ($B_1 = B_2 = 5$) (b) for unified screening ($B = 10$).

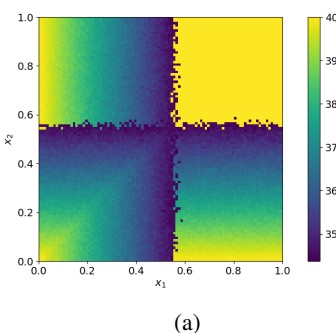
(a)

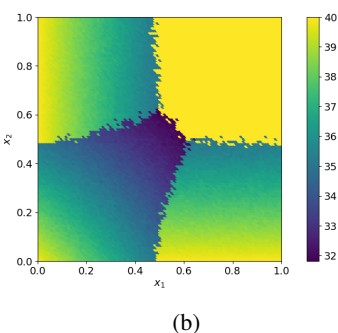
(b)

Figure 19: Survival times ($\mathbb{E}\left[\min\{T_0, T^*\}|a, x\right]$) in (a) independent screening (b) unified screening.

## E. Additional Experimental Details

To assess the robustness of our results, we conduct our Monte Carlo simulation across 50 independent outer iterations. For each iteration, we store the resulting gain and cost estimates and solve a separate instance of the linear program described in Equation (5) using CVXPY to obtain an optimal screening policy. The resulting policies yield average survival times whose standard deviations are $0.0089$ for the unified approach and $0.0117$ for the independent approach. Given that the mean difference in survival time between the two methods is approximately $0.2$, the 95% confidence intervals (mean $\pm$ $2 \times$ std) for each method do not overlap. This provides strong empirical evidence that the observed advantage of unified screening is statistically significant and not due to random variation.

# F. Table of Notations

Table 2: Table of Notations

| Symbol | Description |
|---|---|
| $N$ | Number of diseases |
| $\Theta_n$ | Latent disease state for the $n$th disease, $\Theta_n \in \{0,1\}$ |
| $T_n$ | Time of adverse event for the $n$th disease, $T_n \in \mathbb{N}_+$ |
| $X$ | Patient risk vector, $X \in [0,1]^N$, $X = [X_1, X_2, ..., X_N]^\mathsf{T}$ |
| $X_i$ | Prior probability of having disease $i$, $X_i \in [0,1]$ |
| $Y_n(t)$ | Value of the $n$th screening target at time $t$, $Y_n(t) \in \mathbb{R}$ |
| $\delta_n(t)$ | Screening action for the $n$th target at time $t$, $\delta_n(t) \in \{0,1\}$ |
| $\hat{\theta}_n(t)$ | Diagnostic action for the $n$th disease at time $t$, $\hat{\theta}_n(t) \in \{0,1\}$ |
| $t_n^*$ | Diagnosis time for the $n$th disease |
| $\mathcal{O}_n$ | The event that diagnosis is on time of the $n$th disease, such that $\mathcal{O}_n = \{\Theta_n = 1\} \wedge \{t_n^* < T_n\}$. |
| $\gamma_n$ | Diagnosis threshold for the $n$th disease |
| $c_n$ | Cost per sample from the $n$th screening target |
| $B$ | Budget for screening costs |
| $\alpha_n$ | Maximum allowed false positive rate for the $n$th disease |
| $\mathcal{I}_\delta(T)$ | Information available to the screening policy at time $T$ |
| $\mathcal{I}_{\hat{\theta}}(T)$ | Information available to the diagnostic policy at time $T$ |
| $T^*$ | Time of the first adverse event that is not diagnosed on time |
| $T_0$ | Survival time limit |
| $\rho(X)$ | Referral decision vector based on risk vector $X$, $\rho(X) = [\rho_1(X), \rho_2(X), ..., \rho_N(X)]^\mathsf{T}$ |
| $\bar{\rho}(X)$ | Independent referral decision vector such that $\bar{\rho}(X) = [\bar{\rho}_1(x_1), \bar{\rho}_2(x_2), ..., \bar{\rho}_N(x_n)]^\mathsf{T}$ |
| $a$ | Action index representing referral decisions, $a \in \{1, \ldots, 2^N\}$ |
| $q(a|x)$ | Probability of selecting action $a$ given risk vector $x$ |
| $r_{a,x}$ | Expected survival time given action $a$ and risk $x$ ($\mathbb{E}[\min\{T_0, T^*\}|a, x]$). |
| $\{\tau_{n1}, \ldots, \tau_{nS_n}\}$ | The sampling schedule for the $n$th disease, where $\tau_{ni}$ is the time point that the $i$th sample is taken. |
| $c_n \mathbb{E}[\#_n|a,x]$ | Expected total screening cost for the $n$th disease. |
| $m_{a,x}$ | The total screening cost given action $a$ and $x$ ($\sum_n c_n \mathbb{E}[\#_n|a,x]$). |
| $\lambda$ | Lagrange multiplier for cost constraint. |
| $\kappa_i(x)$ | Parameter defined as $\kappa_i(x) := \mathbb{E}[\min\{T_0, T^*\}|a = i, x] - \lambda \mathbb{E}[\#|a = i, x]$ for the $i$th action and given the vector $x$. |
| $\rho_n^*(x)$ | Optimal referral decision for disease $n$ based on risk vector $x$. |
| $\rho^*(x)$ | Optimal referral decision based on risk vector $x$, $\rho^*(x) = [\rho_1^*(X), \rho_2^*(X), ..., \rho_N^*(X)]^\mathsf{T}$. |

Table 3: Table of Notations (cont.d)

| Symbol | Description |
|---|---|
| $\rho_{i,j}(x)$ | Referral sub-rule to determine $\rho_n^*(x)$ for $N = 2$. |
| $S_{T^*}(t, x, a)$ | Survival function based on $T^*$, such that $S_{T^*}(t, x, a) = \mathbb{P}\{T^* > t | a, x\}$ |
| $F_{T_n}(t)$ | The CDF of the adverse event times $T_n$. |
| $b_\perp(x_2)$ | Boundary function for the first disease's screening sub-rule $\rho_{2,1}(x)$ such that $\rho_{2,1}(x) = \mathbb{1}\{x_1 > b_\perp(x_2)\}$ |
| $b_\equiv(x_2)$ | Boundary function for the first disease's screening sub-rule $\rho_{4,1}(x)$ such that $\rho_{4,1}(x) = \mathbb{1}\{x_1 > b_\equiv(x_2)\}$ |
| $b_\top(x_2)$ | Boundary function for the first disease's screening sub-rule $\rho_{4,3}(x)$ such that $\rho_{4,3}(x) = \mathbb{1}\{x_1 > b_\top(x_2)\}$ |
| $\zeta_1(x_2)$ | Optimal referral decision boundary for the first disease dependent on $x_2$, such that $\rho_1^*(x) = \mathbb{1}\{x_1 > \zeta_1(x_2)\}$. |
| $\zeta_2(x_1)$ | Optimal referral decision boundary for the second disease dependent on $x_1$, such that $\rho_2^*(x) = \mathbb{1}\{x_2 > \zeta_2(x_1)\}$. |
| $\bar{\rho}_n^*(x_n)$ | Optimal independent referral decision for the nth disease that is not dependent on $x_i, \forall i \neq n$, when the referral decisions are restricted to the set of the independent referral decisions $\bar{\rho}_n(x_n)$. |
| $\bar{\zeta}_1$ | Optimal independent referral decision boundary for the first disease that is not dependent on $x_2$, such that $\bar{\rho}_1^*(x_1) = \mathbb{1}\{x_1 > \bar{\zeta}_1\}$. |
| $\bar{\zeta}_2$ | Optimal independent referral decision boundary for the second disease that is not dependent on $x_1$, such that $\bar{\rho}_2^*(x_2) = \mathbb{1}\{x_2 > \bar{\zeta}_2\}$. |

