# OpenReview forum: "Unified Screening for Multiple Diseases"
_ICML.cc/2025/Conference — ICML 2025 poster_

### Official Review · Reviewer_kr4k · 2025-03-11

**Overall Recommendation:** 3

**Summary:**

The problem of screening for multiple diseases is formalized as an optimization problem, specifically for the case where policies for each disease are predefined and the task is to decide which policies to activate given a vector of prior risks. Under a fixed budget and a few simplifying assumptions like sequential test independence, the optimal decision boundaries are studied analytically and through simulations.

**Claims And Evidence:**

The main claims are largely supported. However, the results in this paper are more limited than what is stated in the introduction, and not all of the simplifying assumptions are clearly stated throughout. The referral problem that is ultimately considered has just two diseases with competing risks, policies with deterministic screening schedules, and independent screening tests.

I believe that the language used to describe the method is overly complex, obscuring key design choices while taking up unnecessary space. For instance, the main optimization problem presented in Equation 2 relies on an integer mapping for sets of binary decisions that seems to be a minor implementation detail, and greatly increases the complexity of the notation. It was a challenge to parse through details like these and figure out what is really going on in the core problem presented by this paper.

**Essential References Not Discussed:**

Not to my knowledge.

**Experimental Designs Or Analyses:**

The experimental settings are fully synthetic and rather simple.

**Methods And Evaluation Criteria:**

Unfortunately, the verbosity mentioned above appears to have missed the need to justify key parts of the problem setting and the methodology. Starting from the beginning:
 * Please clarify the role of the screening target $Y_n$ when it is introduced.
 * Why can you assume that screening samples taken at different times are independent? Wouldn't nearby samples be more correlated? Or are these errors entirely due to measurement noise rather than an evolving disease?
 * Why can we assume that we observe the whole $x$ vector? It would be helpful to relate this to the medical examples that are mentioned earlier in the paper.
 * The objective of maximizing the time of the first adverse event that goes undiagnosed needs to be supported. Intuitively, it makes sense, but one could imagine many other objectives that accomplish similar things using expectations rather than minima.

**Other Comments Or Suggestions:**

Line 246: the equation has a typo. The variable $n$ appears to be used for two different purposes at once, $\delta_n$ and also $\exists n$ within the definition.

**Other Strengths And Weaknesses:**

The problem under consideration is highly significant. It would have strengthened this paper to present an actual algorithm for computing decision rules in one of the settings considered. All of the empirical results come from simple synthetic experiments.

**Questions For Authors:**

Could you give some intuition on how the referral optimization task is not equivalent to a simpler policy learning task, with a cost function that takes on a more specific structure?

**Relation To Broader Scientific Literature:**

My main concern relating to the broader literature is that the analysis of the problem might be artificially complicated by all the details that were introduced, and that the decision boundaries possibly correspond to much simpler results from decision theory, especially after you take into account all of the simplifying assumptions that were made along the way.

**Theoretical Claims:**

The proofs for the lemmas appear to be correct, although I did not verify every detail.

---

> ### Author Rebuttal · Authors · 2025-04-01
>
> We thank the reviewer for the thorough review of our paper and constructive comments.
>
> **Limited results:** In (1), we formalize the joint screening and diagnosis problem. This is distinct from (2), which focuses on the referral problem. We choose to solve (2) rather than (1) because our aim is not to propose entirely new screening and diagnosis policies—which can be difficult to justify and implement in clinical practice. Instead, we build upon existing screening policies already endorsed in clinical guidelines.
>
> Characterizing the optimal referral policy becomes significantly more complex and far less intuitive when considering more than two diseases. One would expect the number of boundaries of interest (e.g., $5$ as shown in Figure 1(b) for $N=2$) to grow combinatorial with $N$, as one must account for boundaries that separate screening decisions across different subsets of diseases.
>
> We adopt periodic screening policies with deterministic schedules because such approaches are widely used in clinical practice, as recommended by numerous guidelines—particularly for the early detection of chronic or progressive diseases. Common examples include annual mammograms for breast cancer screening, biennial colonoscopies for colorectal cancer, and regular HbA1c tests for diabetes monitoring.
>
> **Screening target $Y_n$:** May represent a biomarker value, clinical metric, or similar quantity. Conditional independence over time is utilized in likelihood ratio and in proofs characterizing the decision boundaries. While the underlying disease state may evolve over time, this evolution can be modeled as a drift in the expectation of $Y_n$. The randomness in $Y_n$ is then primarily attributed to measurement noise or natural biological fluctuations, which we assume to be independent across time.
>
> **Why $x$ is entirely observable?** One can use off-the-shelf risk prediction models to obtain disease risks for particular diseases. For instance, Gail model (for breast cancer), QRISK3 (for cardiovascular disease), (normalized) polygenic risk scores, and even AI-based models can be used to provide risk scores.
>
> **Why objective make sense:** Our objective can be seen as a simplification of the standard Quality-Adjusted Life Years (QALYs) framework. Each year of life is either $1$ (acceptable quality, e.g., healthy) or $0$ (unacceptable quality, e.g., impaired, suffering). If the disease is detected before the adverse event happens, we assume issues related to the disease can be resolved or managed at a level such that the disease will not cause an unacceptable quality of life. However, after the adverse event happens, the disease causes unacceptable quality of life.
>
> **Relation to broader literature:** To the best of our knowledge, there is no existing decision-making problem in the literature that yields the same structure of decision boundaries as those derived in our work. While our screening cost budget constraint bears some resemblance to the constraint in knapsack problems, the underlying objective in our setting is fundamentally different.
>
> **About empirical results:** We agree that any simulation experiment may not fully capture the real-world benefits of joint screening. A definitive evaluation of our joint screening protocol versus independent screening (current standard practice as indicated by many guidelines) would require a large-scale randomized controlled trial with two arms—an undertaking that is beyond the scope of our current work. Nonetheless, we hope that our theoretical results and in-silico experiments, which demonstrate the potential benefits of joint screening in a simplified yet intuitive and representative setting, will inspire further empirical research in this area.
>
> **About an actual algorithm:** Our referral problem (2) is a linear program (LP) and can be solved by utilizing any LP solver. Some coefficients in this LP are computed by Monte Carlo sampling.
>
> **Relation to policy learning:** If the goal were to learn the optimal referral rule directly from a dataset of patient screening trajectories, the problem could indeed be framed as a policy learning task with an appropriately defined cost function. However, our work takes a complementary, orthogonal approach.
>
> We begin with a well-defined optimization problem (2) and focus on analytically characterizing the structure of its optimal solution (the decision boundaries described in Figure 1(b) and formalized in Proposition 4.5). Rather than learning from data, our emphasis is on understanding the geometry and properties of the optimal referral rule under a known probabilistic model.
>
> Our results suggest a follow-up question: can the characterized decision boundaries be leveraged to develop sample-efficient algorithms for learning optimal referral rules from limited data? We view this as an important direction for future work.
>
> We hope that our response has addressed your concerns. Please let us know if you have any other concerns.

---

> > ### Comment · Reviewer_kr4k · 2025-04-04
> >
> > Thank you for addressing many of my points of concern. I trust that the revised manuscript will reflect at least some of these clarifications.

---

### Official Review · Reviewer_RcdJ · 2025-03-14

**Overall Recommendation:** 3

**Summary:**

This paper proposes a framework for unified screening of multiple diseases under budget constraints and competing risks. The authors formulate this as a referral problem where they choose which screening policies to activate based on patient risk profiles. They characterize optimal decision boundaries for the two-disease case and conduct in-silico experiments to compare against independent screening approaches.

## update after rebuttal

After carefully considering the authors' rebuttal, my assessment remains unchanged. While the authors addressed some technical concerns, the fundamental issue of limited practical applicability persists. The proposed method still lacks validation with real-world medical data, which is crucial for demonstrating clinical relevance and potential impact. For a method targeting medical applications, empirical validation with representative clinical data is essential to establish both reliability and utility.

**Claims And Evidence:**

The core claims are supported by clear and convincing evidence through a combination of rigorous theoretical analysis and comprehensive experimental validation. The limitations are appropriately acknowledged, and the evidence presented aligns well with the scope of claims made. The paper maintains high standards of scientific rigor in both its theoretical and empirical components.

**Essential References Not Discussed:**

NA

**Experimental Designs Or Analyses:**

The experimental design lacks clinical realism and proper statistical rigor is needed to support the practical claims. For example, parameter selection lacks justification (arbitrary choices for budget and screening costs, no explanation for choosing $T_0$ and $\mu_n$ for survival times). For methodology comparison, independent screening baseline uses equal budget split without exploring other allocations. Only minor improvement in survival time (37.70 vs 37.47 years) is observed, which may not be statistically significant, but no statistical analysis provided. Provided that this is on a simulated data, the practical impact of the paper's results appears to be questionable.  The paper would be significantly strengthened by additional comparison to any existing clinical screening protocols.

**Methods And Evaluation Criteria:**

While the theoretical method is sound, the evaluation would be much stronger with real medical datasets and more realistic screening scenarios that reflect actual clinical practice. The current evaluation framework demonstrates mathematical correctness but not practical applicability.

**Other Comments Or Suggestions:**

NA

**Other Strengths And Weaknesses:**

Strengths:
- The mathematical characterization of optimal policies is rigorous and well-developed.

Weaknesses:
1. Limited practical applicability:
- Only handles 2 diseases
- Relies on simplified assumptions about screening schedules
- No validation with real-world medical data
- Doesn't address clinical implementation challenges

2. Minor improvements:
- Small gain in survival times (37.70 vs 37.47 years) may not justify the increased complexity
- No cost-benefit analysis of implementation effort versus expected gains

3. Missing context:
- Limited discussion of how this would integrate with existing clinical protocols
- No consideration of practical constraints like patient preferences or healthcare system limitations
- Doesn't address how to handle more realistic scenarios with uncertain disease risks

The paper makes solid theoretical contributions but needs stronger connections to clinical practice to demonstrate real-world significance.

**Questions For Authors:**

1. Figure 1 interpretation:
- Could you explain the meaning of the curved boundaries in Figure 1(b)? What determines their shape and why do they differ from the straight-line boundaries in 1(a)?
- What is the practical interpretation of the different colored regions in terms of clinical decision-making?
A clear explanation would help evaluate the practical relevance of your theoretical results.

2. Scaling to more diseases:
- What are the theoretical and computational challenges in extending your approach to 3+ diseases?
- Do you expect qualitatively different behavior in the decision boundaries with more diseases?
- How would the computational complexity scale?

3. Clinical validation:
- Have you validated any aspects of your model against real clinical screening data or protocols?
- What modifications would be needed to handle real-world factors like uncertain disease risks and variable screening costs?
This would help evaluate whether the theoretical gains would translate to practice.

4. Statistical significance:
- Did you perform statistical analysis on the survival time improvement (37.70 vs 37.47 years)?
- How sensitive are these results to your parameter choices and assumptions?
Understanding the robustness of the improvements would affect assessment of the paper's impact.

**Relation To Broader Scientific Literature:**

While prior work like Wright et al. (2015) and Peng & Xiang (2021) studied competing risks in single-disease contexts, this paper extends the framework to optimize screening decisions across multiple diseases simultaneously. However, there are some limitations in connecting to literature.

- Does not fully explain how their optimal policy compares to existing clinical guidelines.
- Limited discussion of how their theoretical results relate to real-world screening scenarios.
- No comparison to other optimization approaches in healthcare resource allocation.

In summary, while the paper builds on existing ideas in disease screening and risk modeling, it could better contextualize its theoretical advances against practical clinical approaches in the field.

**Theoretical Claims:**

I reviewed some of the theoretical proofs, focusing primarily on Section 4 which contains the key theoretical claims about optimal screening policies. Other results seem to be correct but I have not checked the detailed proofs in the appendix.  Additional verification by other reviewers would be valuable.

---

> ### Author Rebuttal · Authors · 2025-04-01
>
> Thank you for the thorough review of our paper and constructive comments.
>
> **Figure 1** 1(a) shows the decision boundaries for independent screening (current standard). 2(a) shows the boundaries that characterize the optimal policy for our referral problem (2). Our main contribution is to mathematically characterize these boundaries (Lemma 4.1 to Lemma 4.4, Proposition 4.5). Main message: to act optimally, screening for disease 1 should also depend on the risk of disease 2. Boundary shapes also depend on the screening budget, screening costs, adverse event times, and risk distribution (see Appendix C). Practical interpretation: Clinics can use 1(b) to decide for what a patient with risk vector $(x_1,x_2)$ should be screened.
>
> **Scaling to more diseases:** $>2$ diseases does not put significant computational burden. Our referral policy optimization problem is a linear program (LP); hence, SoTA complexity bounds for LP apply to our setting.
>
> Characterization of the optimal referral policy will be much more complex and far less intuitive for the case of $>2$ diseases. One would expect the number of boundaries of interest (e.g., $5$ as shown in Figure 1(b) for $N=2$) to grow combinatorially with $N$ as there can be boundaries separating screening of one subset of diseases from another.
>
> **Clinical validation:** Any simulation study will not fully capture the real-world benefits of joint screening. A definitive evaluation of our joint screening protocol versus independent screening (current standard) would require a large-scale randomized controlled trial with two arms—an undertaking that is beyond the scope of our current work. Nonetheless, we hope that our theoretical results and in-silico experiments, which demonstrate the potential benefits of joint screening in a simplified yet intuitive and representative setting, will inspire further empirical research in this area.
>
> **Uncertain disease risks and variable screening costs:** One can use off-the-shelf risk prediction models to obtain disease risks for particular diseases. For instance, Gail model (for breast cancer), QRISK3 (for cardiovascular disease), (normalized) polygenic risk scores, and even AI-based models can be used to provide risk scores. In our formulation, screening costs for different diseases can be different. In Figure 4, we also show the effects of varying screening costs.
>
> **Statistical significance:** Since our experiments are on simulated data, we report exact gains obtained by solving (2). Significance tests are not necessary since we are not making claims based on limited data.
>
> **Parameter selection and sensitivity of our results:** The choice of $T_0$ and $\mu_n$ is done by examining clinical literature (first paragraph of Section 5.1). While our choices of screening budget and costs are not borrowed from a real-world study, in Appendix C, we discuss in detail how the behavior and the performance of the optimal referral policy changes as we vary each of these parameters. We hope that our detailed appendix resolves the reviewer's concerns about parameters.
>
> **Limited practical applicability:** There are many other examples where this methodology could be used to guide policymakers. One example can be found in sexual health services. For many settings, human immunodeficiency virus (HIV) and human papillomavirus (HPV) screening protocols are conducted at different health services and hence are independently considered (e.g.,
> https://www.who.int/publications/i/item/9789240024168, https://pubmed.ncbi.nlm.nih.gov/38297406).
>
> Most of these screening protocols are guided by a certain level of risk (e.g., sexual behaviors). These screening measures carry varying costs (HIV antibody is not costly, whereas HPV requires more costs associated with invasive procedures carried out by a gynecologist (for cervical neoplasia) or proctologist (for anal neoplasia)). Finally, delayed screening has implications on the time to present with severe HIV disease (for HIV) and time to develop CIN2/3 or AIN2/3, that is, cervical or anal pre-cancer (for HPV). This methodology could help define which risk levels would warrant joint screening protocols (to plead for streamlined services) or if screening protocols can remain part to minimize these times to undesirable outcomes.
>
> **Minor improvements:** Compute complexity is not an issue, as the problem is LP. For $N=2$, our characterization of the decision boundaries and visualizations (e.g., Fig 1(b)) provide interpretable explanations of who should be screened. The same screening resources can be used more efficiently (no increase in screening costs for the population). We improve over independent screening in all cases (even when disease risks are independent). Reported numbers may vary based on the experimental setup, but the main message remains the same.
>
> We hope that our response has addressed your concerns. Please let us know if you have any other concerns.

---

> > ### Comment · Reviewer_RcdJ · 2025-04-05
> >
> > I appreciate the authors' response, but I remain concerned about two methodological aspects:
> >
> > 1. **Lack of real data validation**: While a full RCT may be beyond scope, retrospective analysis using existing clinical datasets would significantly strengthen your claims. Simulation studies alone, no matter how well-designed, cannot fully capture the complexities of real-world clinical settings.
> >
> > 2. **Absence of statistical testing**: The statement that significance tests are "not necessary" for simulated data mischaracterizes good methodological practice. The reported 0.23 improvement in average survival time requires statistical validation across multiple simulation runs with different seeds to demonstrate that this difference is reliable and not due to chance variation in your specific simulation instance. Without confidence intervals or p-values, it's difficult to interpret the practical significance of this finding.
> >
> > These additions would substantially strengthen the paper's conclusions and enhance its potential impact on clinical practice.

---

> > > ### Author Response · Authors · 2025-04-08
> > >
> > > We appreciate the reviewer’s emphasis on the need for statistical validation and would like to clarify that our results are not based on a single simulation. At each Monte Carlo iteration, we simultaneously run 200 independent simulations for each 10000 (x1, x2) pair to capture the variability in patient trajectories, diagnoses, and event occurrences. This inner-loop simulation ensures that the survival outcomes reflect a realistic distribution over possible patient outcomes, rather than a single realization.
> > >
> > > To further validate the robustness of our findings, we conducted an experiment where we saved the outputs of each of the 50 Monte Carlo iterations separately. Using this data, we solved 50 different linear programs to obtain optimal policies and corresponding average survival times for both the unified and independent screening approaches.
> > >
> > > The results across these runs show that the standard deviation of the average survival time is 0.0089 for unified screening and 0.0117 for independent screening. Given that the average difference in survival time between the two methods is approximately 0.2 (as shown in our work), the confidence intervals defined as mean ± 2 * std do not overlap. This provides strong empirical evidence that the observed improvement is not due to chance variation but reflects a consistent advantage of our unified approach.

---

### Official Review · Reviewer_yUUd · 2025-03-14

**Overall Recommendation:** 3

**Summary:**

This article offers a novel optimization framework for the complex task of engaging in unified screening for multiple diseases. They offer a novel optimization framework, attempting to balance multiple factors including disease risk, budget, and diagnostic test characteristics.

**Claims And Evidence:**

The authors begin with a reasonable summary of the challenges of disease screening and the optimal use of resources in this context, although this could be clarified with respect to why we might not just stack these individual screening programs in all patients (which is implicit but may benefit from being made explicit). It is not entirely clear to me why the screening for one disease depends on the screening for another, outside of contexts where the risks are clearly contingent (e.g. pulmonary hypertension and cardiac disease). Is this approach meant to be limited to such contexts?

They further offer an in silico evaluation, attempting to assess the impact of their various policies in a simulated environment. While the in silico analysis is reasonably performed within these boundaries, I worry that the overall approach overstates the benefit of many screening programs implicitly. While the benefit of screening is intuitive, it often does not play out in practice (this excellent recent review should be discussed, as most screening programs have not actually been shown to offer any significant mortality benefit https://pmc.ncbi.nlm.nih.gov/articles/PMC10463170/). This should be softened.

The authors claim "For example, treating one condition, such as heart disease, can enhance the effectiveness of screening for another, such as lung cancer" without citation. What is meant by this? Is this referring to improving the accuracy of e.g. a chest x-ray because  there is less pulmonary edema?

Overall, however, despite some of my concerns about the broader direction of the field (and my deeper skepticism of screening as naively applied), I feel that this paper makes a reasonable contribution to its literature field.

I do, however, believe that these results must be discussed more cautiously, with the recognition that these toy formulations of screening may not fully align with the complex realities of clinical medicine in this context. Similarly, I feel the authors should provide some further justification of the value of this multi-disease screening approach and the specifics of their contentions regarding the relatedness of these diseases.

**Essential References Not Discussed:**

As discussed above, further engagement with some of the medical literature skeptical of the broader benefits of screening may be worthwhile here.

**Experimental Designs Or Analyses:**

I am not able to review the mathematical proofs in detail given my background, however their in silico analyses are reasonable overall.

**Methods And Evaluation Criteria:**

The authors offer a detailed mathematical optimization formula to combine multiple different policies. It is commendable that their methods attempt to incorporate multiple different areas of analysis.

I am not able to comment in detail on the accuracy of their formulae, as my background is more in clinical medicine and practical applications of machine learning in clinical contexts.

**Other Comments Or Suggestions:**

One quibble with the introduction is with the statement "For example, screening a patient with high risks for both lung cancer and cardiovascular disease for only one condition might fail to improve their overall health outcomes, whereas a unified screening approach could yield better results". This does not make it clear why one would not just screen for both in this patient. I believe it is substantiated elsewhere, bus this should be more clearly explained in the introduction.

**Other Strengths And Weaknesses:**

Discussed elsewhere.

**Questions For Authors:**

See above - I have several questions regarding the underlying theoretical assertion of the connection between the multiple diseases being screened.

**Relation To Broader Scientific Literature:**

The authors offer an excellent engagement with the relevant literature, with appropriate commentary in the related works section. I also appreciate the excellent Table 1 approach to clearly situating this project within the broader literature. My concerns as outlined above are with a lack of engagement with literature skeptical of screening overall.

**Theoretical Claims:**

As outlined above, I worry that this work (as with much of the work in this field) overstates the effectiveness of screening in general.

---

> ### Author Rebuttal · Authors · 2025-04-01
>
> We thank the reviewer for the thorough review of our paper and the constructive comments.
>
> **Why screening of one disease depends on the screening of another:** In the example of pulmonary hypertension and cardiac disease, risks are clearly contingent. Our approach is not limited to such examples, as we offer a non-trivial solution even when disease risks are independent. Due to the nature of our objective function (related to survival time) and our constraint function (screening cost budget), the optimal threshold for activating the screening of one disease depends on the risk of the other disease. This coupling distinguishes our model and solution from independent screening, which yields a suboptimal policy for our referral problem. We will further clarify this in the revised paper.
>
> **Regarding the benefit of screening:** Thanks for pointing out this important review paper. We agree that screening does not always offer benefits and sometimes can even be harmful, as in the case of overdiagnosis. We will mention this review paper in the revised version and explicitly discuss the limitations of screening in line with this work. In our work, we focus on the case when screening does not hurt. In our case, the only detrimental effect of screening is a false positive rate, which is expressed as a constraint in the optimization problem. We cannot screen everyone for every disease since screening is costly and screening resources are limited. We characterize exactly how limited screening resources should be distributed over the population so that the expected benefit is maximized.
>
> **Clarification of the screening example:** Consider a patient who (potentially) has both heart disease and cancer but is only screened for cancer, where the adverse event from cancer is expected to occur after the adverse event from heart disease. If we only screen for cancer, but the patient unexpectedly dies from heart disease, then the cancer screening offers no benefit to the patient in terms of lifetime gain. If the patient is screened for both diseases, heart disease will be identified earlier, the adverse event due to heart disease will be prevented, and cancer screening will result in lifetime gain by detecting cancer before the adverse event associated with it happens.
>
> We hope that our response has addressed your concerns. Please let us know if you have any other concerns.

---

### Decision · Program_Chairs · 2025-05-01

**Decision:**

Accept (poster)

**Comment:**

This paper describes a novel optimization framework for unified screening of multiple diseases. All three reviewers gave weakly positive ratings, but they raised several important concerns. While the problem framework is novel, the resulting claims are sound, and the in-silico experiments are well-reasoned, the paper itself lacks grounding in actual clinical motivation. The lack of real-world data and relationship to existing clinical guidelines (raised by RcdJ) raise concerns about the overall contribution of this work for clinical science, which it claims as the main motivation for the paper. The paper only focuses on 2 diseases for screening, and there were concerns that the work appears to not consider the potentially detrimental effects of screening (yUUd), it does not explain how the optimal policy compares to existing clinical guidelines (RcdJ), and the presentation of material is overly unclear (kr4k).

With the acknowledgement of the limitations, I still find the framing of the paper interesting, and the theoretical and in-silico claims are sound. For this reason, I'm recommending a weak accept.